# Respiratory symptoms and pulmonary function in paint industry workers exposed to volatile organic compounds: A systematic review and meta-analysis

**Lavanya Sekhar** [1], **Akila Govindarajan Venguidesvarane**[2], **Gayathri Thiruvengadam**[3], **Yogita Sharma**[4], **Vidhya Venugopal**[5], **Santhanam Rengarajan**[6], **Priscilla Johnson**[1] *

1 Department of Physiology, Sri Ramachandra Medical College & Research Institute, Sri Ramachandra Institute of Higher Education and Research(SRIHER), Chennai, Tamil Nadu, India, 2 Department of Community Medicine, Sri Ramachandra Medical College and Research Institute, Sri Ramachandra Institute of Higher Education and Research (SRIHER), Chennai, Tamil Nadu, India, 3 Faculty of Allied Health Sciences, Sri Ramachandra Institute of Higher Education and Research (SRIHER), Chennai, Tamil Nadu, India, 4 Department of Mathematics and Statistics, University of Victoria, Victoria, British Columbia, Canada, 5 Department of Environmental Health Engineering, Faculty of Public Health, SRIHER (DU), Chennai, Tamil Nadu, India, 6 Department of Neurosurgery, Sree Balaji Medical College and Hospital, Bharath Institute of Higher Education and Research (BIHER), Chennai, Tamil Nadu, India

☯ These authors contributed equally to this work.
* priscillajohnson@sriramachandra.edu.in

## Abstract

Several epidemiological studies have examined the respiratory consequences of occupational exposure to volatile organic compounds (VOCs). However, their effects on paint industry workers in organised and unorganised occupational sectors vary. The present systematic review and meta-analysis aim at evaluating the respiratory symptoms and pulmonary function of paint industry workers from various occupational sectors exposed to VOCs. Relevant MESH terms were used for literature search in MEDLINE, Scopus, Web of Science, and Google Scholar till August 2023. The articles were independently retrieved and qualified by two reviewers and two subject experts arbitrated reviewer differences to establish relevant article inclusion. The systematic review comprised 23 observational studies that assessed respiratory symptom and pulmonary function tests (PFT) among paint industry worker from various occupational sectors. The meta-analysis included 12 studies on respiratory symptoms and 18 on PFT. Pooled meta-analysis was done using random effect model, and the crude odds of respiratory symptoms such as cough (OR: 2.72, 95% confidence interval [CI]: 1.74 to 4.25), dyspnoea (OR: 3.59, 95% CI: 2.13 to 6.05), nasal/throat irritation (OR: 4.5, 95% CI: 1.7 to 12.1), and wheezing (OR: 2.28, 95% CI: 1.37 to 3.82) were significantly higher among paint industry workers exposed to VOC compared to unexposed population. PFT parameters, such as forced expiratory volume in one second (FEV1) (SMD: -0.88, 95% CI: -1.5 to -0.2) and FEV1/forced vital capacity (FEV1/FVC) (SMD: -0.97, 95% CI: -1.6 to -0.32) were found to be significantly reduced among the paint industry workers. The meta-analysis has helped in generating evidence regarding the effect of VOC on respiratory symptoms and pulmonary function and the strength of the association varied

**Data Availability Statement:** All relevant data are within the article and its Supporting Information files.

**Funding:** The author(s) received no specific funding for this work.

**Competing interests:** The authors have declared that no competing interests exist.

with geographical regions, and the type of occupational sectors. Despite the heterogeneity ($I^2 > 75\%$) of studies, statistical power of this analysis was significant.

**Trial registration**: PROSPERO registration number: CRD42022311390.

## Introduction

Workplace exposure to volatile organic compounds (VOCs) has a detrimental effect on human health and is a major public health challenge among workers in various occupational sectors worldwide [1, 2]. VOCs are one of the major indoor air pollutants and represent a broad spectrum of organic compounds such as benzene, ethyl benzene, toluene, xylene, styrene, N-butyl acetate, isobutyl acetate, acetone, ethanol, etc. These molecules have a high vapour pressure at room temperature that are emitted from diverse chemical sources in everyday use, like solvents, paints, cleaning and degreasing agents, pesticides, and personal care products [3]. Since a large number of these compounds evaporate or sublimate in the surrounding air, workplace exposure to VOCs exerts a negative effect on human health, ranging from short-term effects, such as mild irritation of the eyes, nose, throat, or skin, headache, nausea, and asthma-like symptoms, to long-term complications, such as restrictive or obstructive disorders like chronic obstructive lung disease (COPD) [2, 4]. Several studies have shown that VOCs, which are commonly found in paints, lacquers, and thinners, have a dose-related negative impact on upper and lower respiratory symptoms [5–7]. Since the VOCs emitted from paints have a rapid 'off-gassing' effect, painters working in various organised and unorganised occupational sectors are more vulnerable to VOC exposure and its adverse effects [8].

Epidemiological studies among paint industry workers have reported respiratory symptoms, such as cough, dyspnoea, chest tightness, wheezing and reduced pulmonary function in both restrictive and obstructive dysfunctions. This association was influenced by the duration and intensity of exposure [9–12]. Nevertheless, the role of confounding factors, such as age, geographical areas, ethnicity, smoking status, occupational sectors have to be considered. Therefore, the present systematic review and meta-analysis was conducted to summarize information on the respiratory symptoms and pulmonary function among paint industry workers in both organised and unorganised sectors exposed to VOCs present in paints.

## Materials and methods

The current systematic review was carried out in accordance with the Meta-analysis Of Observational Studies in Epidemiology (MOOSE) guidelines [13] (S1 Table).

### Data sources and search strategy

A comprehensive search strategy was applied to retrieve relevant literature published from inception to August 2023 from PubMed, Scopus, and Web of Science databases. Following the preliminary search, appropriate keywords (MeSH Terms) were formulated (Table 1).The articles were sought using boolean operators. The review was registered in the International Prospective Register of Systematic Reviews (PROSPERO) (Reg ID: CRD42022311390).

A few peer-reviewed and non-indexed articles were not captured in the above mentioned databases. Therefore, the literature search was extended using Google Scholar and a grey search procedure in the Google search engine using similar search terms (Table 1 and Fig 1).

**Table 1. Search terms for the literature review.**

| Search Strategy | Search terms |
|---|---|
| **Population** | paint shop workers, house painters, construction painters, lacquer workers, automobile painters, spray painters, paint industry workers |
| **Exposure** | volatile organic compounds, organic solvents, paints, lacquer coatings, solvents |
| **Outcome** | respiratory functions, lung functions, airway symptoms, respiratory symptoms, rhinitis, respiratory tract diseases, lung diseases, cough, wheezing, dyspnoea, throat irritation, pulmonary function |

## Study selection

**Inclusion criteria.** Studies were included if they met the following criteria: (i) articles published in English language (ii) observational studies (iii) studies involving painters working in industrial sectors within the age group of 15 and 60 years, exposed to VOCs present in paints, thinners, and lacquers, (iv) studies assessing respiratory symptoms such as cough, dyspnoea, nasal or throat irritation, wheezing and (v) studies evaluating pulmonary function parameters (forced expiratory volume in the first second [FEV1], forced vital capacity [FVC], and FEV1/FVC) assessed using spirometry.

**Exclusion criteria.** Studies were excluded if they were (i) animal studies and experimental models (ii) devoid of control groups (iii) dealing with major exposures other than VOCs present in paints; and (iv) outcomes other than respiratory symptoms or pulmonary function.

**PRISMA 2020 flow diagram for new systematic reviews which included searches of databases, registers and other sources**

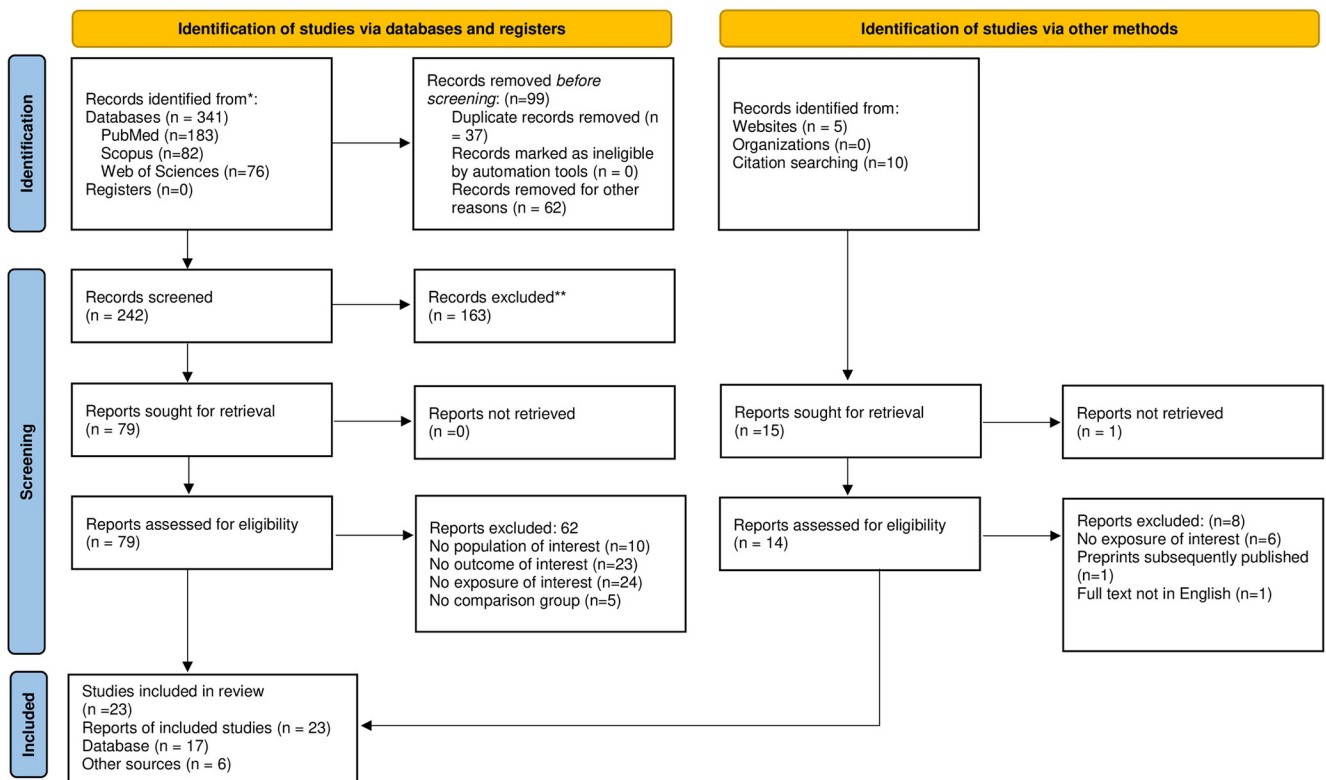

*Consider, if feasible to do so, reporting the number of records identified from each database or register searched (rather than the total number across all databases/registers).
**If automation tools were used, indicate how many records were excluded by a human and how many were excluded by automation tools.

*From:* Page MJ, McKenzie JE, Bossuyt PM, Boutron I, Hoffmann TC, Mulrow CD, et al. The PRISMA 2020 statement: an updated guideline for reporting systematic reviews. BMJ 2021;372:n71. doi: 10.1136/bmj.n71. For more information, visit: http://www.prisma-statement.org/

**Fig 1. Flow diagram reporting article selection process.**

**Data abstraction.** The MOOSE reporting guidelines were used for the selection of articles and data synthesis. The resulting citations which included the records identified through database search and other sources after initial filtration (n = 356) were stored in the Zotero reference manager. Duplicates (n = 37) were removed using the same. A two-stage screening procedure was used to identify the primary articles. Two reviewers independently screened the titles (n = 319) and abstracts of the resulting citations using the inclusion and exclusion criteria. After screening the titles and abstracts, full-text screening (n = 94) was performed, and potentially relevant studies which assessed the pulmonary function and respiratory symptoms among paint industry workers exposed to VOC were included in the systematic review (n = 23) (Fig 1). The reviewers independently extracted and recorded data from the twenty three articles on Google sheets. The following information was extracted from each study: author's names, year of publication, study design, study setting, total number of exposed (sample size) and unexposed subjects, age, years of exposure, outcome parameters, type of occupational sector, and their findings. Among the screened studies, four studies reported respiratory symptoms solely, nine studies assessed—Pulmonary Function Tests (PFT), while ten studies reported both PFT and respiratory symptoms. Two among the ten studies that assessed both respiratory symptoms and PFT were excluded from the meta-analysis of respiratory symptoms because they failed to compare respiratory symptoms in the control group [12, 14]. Nevertheless, these studies were incorporated in the PFT meta-analysis. Similarly, one study [15] was excluded from the PFT meta- analysis due to lack of adequate information about the PFT parameters of the control group. However, it was incorporated in the meta-analysis of respiratory symptoms. Overall, the meta-analysis incorporated research studies that evaluated respiratory symptoms (n = 12), and pulmonary function (n = 18). If any study lacked sufficient information, the authors of the original article were contacted via email to obtain further information or clarification. Any disagreements at any stage between the two reviewers were referred to another two reviewers, who were subject experts, for resolution and to decision making for inclusion of the article in the systematic review and subsequently for the quantitative synthesis.

**Assessment of study quality.** The quality of the studies were assessed based on the modified Newcastle-Ottawa scale for observational studies. The appropriateness of the study design, sample size justification, representativeness, recruitment strategy, response rate, ascertainment of exposure, comparability, generalisability, objectivity/reliability of outcome determination, and appropriate statistical analyses were rated from 0 to 10 (S2 Table) for cross sectional studies [16, 17]. Studies that received a score of >7 were considered good-quality studies with minimal risk of bias and those that scored ≤4 were considered to have a very high risk of bias.

**Data synthesis and analysis.** Meta-analysis was carried out using R (version 4.0.1) software, Metafor package (version 3.4.0) [18] and Stata software version 17.

Descriptive data was reported as combined mean and standard deviation. Data from studies reporting respiratory symptoms were summarised as count data and their crude odds ratios were calculated. A standard mean difference (SMD) was computed for studies that reported PFT parameters, as some of them reported absolute FEV1 or FVC or the ratio, while others reported percentage of predicted values. Consequently, it was not feasible to conduct a direct comparison, necessitating the calculation of SMD. A SMD of 0 and crude odds ratio of 1 was considered null. Random effects model (RE model) was applied to fit the data as the studies showed heterogeneity. The heterogeneity (i.e., $\tau^2$) was estimated using the restricted maximum-likelihood estimator [18]. In addition to the estimate of $\tau^2$, the $Q$-test for heterogeneity [19] and $I^2$ statistics [20] were estimated. In case heterogeneity was detected (i.e., $\tau^2 > 0$, $I^2 > 50\%$ regardless of the results of the $Q$-test), studentized residuals and Cook's distances were used to determine if the studies were outliers or influential in the context of the model.

Studies with a studentized residual value larger than the $100 \, (1-0.05/[2k])^{th}$ percentile of standard normal distribution were considered potential outliers (i.e., using Bonferroni correction with two-sided $\alpha = 0.05$ for k studies included in the meta-analysis). Studies with a Cook's distance larger than the median plus six times the interquartile range of Cook's distance were influential. Further, the studies considered as outliers or influential were restricted and their true effect sizes were reassessed. Publication bias was checked using a regression test, and the standard error of the observed outcomes was used as a predictor to check for funnel plot asymmetry. Moreover, publication bias was also addressed by performing File-drawer (failsafe N—Rosenberg) analysis [21]; a small fail-safe N suggests potential publication bias (fail-safe N <5n+10).

## Results

A total of twenty three cross-sectional studies that met the inclusion criteria were included in the present systematic review. The main characteristics of the studies included are summarised in Table 2.

### Study characteristics

The chosen observational studies were cross-sectional in design and were published from inception to August 2023 across 14 countries worldwide. According to the study quality assessment using a modified Newcastle Ottawa scale (NOS) for observational studies [16, 17], eight studies were considered to be of good quality with minimal risk of bias (NOS:≥7), with the highest score being '8', and the remaining fifteen studies were considered to be satisfactory with the lowest score being '5'(S2 Table). Ten studies were conducted in Africa [Algeria (n = 1) [26]; Egypt (n = 5) [14, 28, 29, 31, 35]; Nigeria(n = 3) [11, 32, 33]; Tanzania (n = 1) [15]]; Eight in Asia [Iraq (n = 2) [25, 27];Turkey (n = 1) [23]; India (n = 2) [12, 34]; Korea (n = 1) [36]; Iran (n = 1) [37]; Pakistan (n = 1) [9]]; and five in Europe and United States of America (USA) [Europe (n = 3) [22, 25, 27] and USA (n = 2) [10, 24]].Thirteen studies were performed among painters in industrial sectors who work in automobile industries, painting workshops, wood and lacquering workshops, and shipyard, three studies were conducted in workers employed in the paint manufacturing industry and seven studies involved construction painters (Table 2). Twelve studies utilized occupational history questionnaire to evaluate VOC exposures, five studies employed ambient air sampling techniques, and one study deployed exposure index assessments.

### Quantitative synthesis

**Effect of VOCs on respiratory symptoms among painters.** For the meta-analysis twelve of fourteen observational studies on respiratory symptoms were considered as depicted in Table 3. Two studies were excluded from quantitative synthesis since they did not report the respiratory symptoms of the control group [12, 14].

The sample size included a total of 1954 participants who had been exposed to organic compounds present in paints and 2506 unexposed participants. The mean age of the exposed and unexposed participants was 40.3±12.5 and 41.5± 12 years, respectively. The average duration of exposure to VOCs was 20.5±15.8 years. Study participants in all the studies were male except for two studies which included both genders [15, 24].

Forest plots showing the observed outcomes and estimates based on the random-effects model for respiratory symptoms, such as cough, dyspnoea, nasal irritation, and wheezing were constructed and are depicted in Figs 2–5.

Table 2. Characteristics of studies included in the systematic review (N = 23).

| SI No | Author & Year of Publication* | Country | Population (Exposed/ Unexposed) | Sample size (Exposed /Unexposed) | Age (Mean ±SD) (Exposed /Unexposed) | Major Exposures & their constituents | Exposure Assessment Methods | Mean Exposure years | Respiratory Symptoms | PFT | Quality |
|---|---|---|---|---|---|---|---|---|---|---|---|
| | Schwartz et al.,1988 [10] | USA | Construction painters/Sheet Metal Workers | 118/314 | 42.3 ± 11.8 / 46.9 ± 11.5 | Spray paint & solvent fumes | Occupational exposure history questionnaire | 15.59 ± 6.87 | Painters reported cough (p<0.05), wheezing (p<0.001), and dyspnoea (p<0.001) | Painters showed reduced FEV1% (p<0.025) & FEV1/FVC (p<0.05). | 6 |
| | Alexandersson et al., 1988 [22] | Sweden | Industrial painters/ matched population control | 38/18 | 34 ±10 / 37±9 | Acid hardening lacquers: n–Butanol, Iso–Butanol, Butylacetate, Ethanol, Etylacetate, Toluene, Xylene &Formaldehyde | Air samplers in respiratory zones | 7.8 | Nasal/throat irritation higher among painters (p<0.01), dyspnoea at work & chest oppression were observed but not significant. | No significant changes reported. No correlation of exposures or shift duration in FVC and FEV1 values. | 7 |
| | Eifan et al., 2005 [23] | Turkey | Automobile painters/ Students | 72/72 | 17.47 ± 0.14 / 17.24 ± 0.14 | Spray paint: solvent fumes | Occupational exposure history questionnaire | 3.12±0.2 | 50% of the painters showed work–related asthma–like symptoms [OR (95% CI)–2.9 (1.026, 8.13)] | No significant difference in PFT parameters. | 7 |
| | Hammond et al., 2005 [24] | USA | Automobile painters / assembly workers | 116/357 | 39.6 /36.2 | Paints: acetone, isopropanol, toluene, xylene, naphtha, hexane, cellosolve acetate, and ethanol | Occupational exposure history and reviews from industrial hygiene records | - | Adjusted odds of cough [1.55 (0.96, 2.53)], asthma symptoms [1.77 (0.87, 3.62)] were higher among painters, but were not statistically significant. Physician-diagnosed odds of COPD (OR: 3.73, 95% CI: 1.27, 11.0) were significantly more among painters. | | 8 |

*(Continued)*

Table 2. (Continued)

| SI No | Author & Year of Publication* | Country | Population (Exposed/Unexposed) | Sample size (Exposed/Unexposed) | Age (Mean ±SD) (Exposed/Unexposed) | Major Exposures & their constituents | Exposure Assessment Methods | Mean Exposure years | Outcome assessed Respiratory Symptoms | Outcome assessed PFT | Quality |
|---|---|---|---|---|---|---|---|---|---|---|---|
| | Kaukiainen et al., 2005 [25] | Finland | Construction painters/ Carpenters | 523/505 | 48.4 ± 8.9 / 49.3 ± 7.9 | Paints: Solvent-based alkyd paints, glues, epoxy/ urethane paints, ecological paints, water-based paints, glues, putties/ plasters | Occupational exposure history questionnaire | 38.06 ±12.27 | Painters reported higher odds of cough [1.69 (1.35, 2.37)], dyspnoea [1.76 (1.12–2.75)], rhinitis [1.66 (1.27–2.17)], and laryngeal symptoms [1.56 (1.04–2.32)]. Chronic bronchitis [OR (95%CI): 2.2 (1.2–4.0)] was associated exposure duration. | | 6 |
| | Ould-Kadi et al., 2007 [26] | Algeria | Paint industry workers/ matched population control | 106/123 | 39.3 ± 6.4 / 38.3 ± 8.3 | Paints & solvents: xylene, toluene, white-spirit, ethyleneglycolacetate, methyl isobutyl ketone and butanol | Occupational exposure history questionnaire | 11.9 ± 4.9 | Workers exposed to organic solvent had three times higher odds of developing [OR (95%CI) - 3.43 (1.09–11.6); p = 0.037] bronchial hyper responsiveness compared to controls. | FEV1%, was 5.6% lower among subjects exposed to solvents (CI: -7.9 to -3.3; p = 0.0009) compared to controls. Other PFT parameters did not show any significant difference. | 6 |
| | Kaukiainen et al., 2008 [27] | Finland | Construction painters/ Carpenters | 288/505 | 48.5 ± 8.8 / 48.9 ± 8.1 | Paints: Solvent-based alkyd paints, glues, epoxy/ urethane paints, ecological paints, water-based paints, glues, putties/ plasters | Occupational exposure history questionnaire | 24.2 ±10.1 | Outdoor painters had increased odds of rhinitis [2.4 (1.1–5.2)], asthma [4.7 (1.4–16.1)], chronic bronchitis [2.9 (1.0–8.4)] compared to indoor painters or carpentry workers. | | 5 |

*(Continued)*

**Table 2.** (Continued)

| SI No | Author & Year of Publication* | Country | Population (Exposed/ Unexposed) | Sample size (Exposed /Unexposed) | Age (Mean ±SD) (Exposed /Unexposed) | Major Exposures & their constituents | Exposure Assessment Methods | Mean Exposure years | Outcome assessed — Respiratory Symptoms | Outcome assessed — PFT | Quality |
|---|---|---|---|---|---|---|---|---|---|---|---|
| | El Mahdy et al., 2009 [28] | Egypt | Industrial painters/ administrative staffs | 38/30 | 40.74±1.63/ 48.33 ±1.85 | Paints & organic solvents: toluene | Air sampling & Urinary ortho-cresol levels. | 17.3 ± 6.9 | | Significant decrease in parameters like FEV1/ FVC% (p = 0.045),FEF 50% (p = 0.005) & MVV % (p = 0.008) among the painters | 7 |
| | Metwally et al., 2012 [29] | Egypt | Paint industry workers/ administrative staffs | 191/182 | 43.28 ±11.43/ 42.29±10.39 | Paints, resins, additives & pigments: ethyl alcohol, ethylene glycol, ethyl glycol acetate, butyl acetate, xylene, toluene, styrene, & other aromatic solvents | Not specified | 18.5±11.04 | Prevalence of respiratory symptoms were significantly higher (p<0.05) among the exposed groups and is high among high exposure group (HEG) | FVC% (p<0.05), FEV1% (p<0.01), FEF 25–75% (p<0.01) were significantly lower among painters. PFT showed a weak negative correlation (r = -0.2; p<0.05) with age and exposure duration. | 5 |
| | Numan 2012 [30] | Iraq | Automobile painters/ matched population controls | 30/30 | 37.9±8.89/ age matched | Spray paints & organic solvents: toluene, xylene, isopropanol | Not specified | 13.39±6.85 | | Significant decrease (p<0.05) in PFT parameters (FVC, FEV1, FEV1/FVC) among the painters. No significant correlation was found between PFT with age and duration of exposure. | 5 |
| | El-Gharabawy et al., 2013 [31] | Egypt | House painters/ matched population controls | 30/30 | 32.27±5.89/ 31.73±6.43 | Mixed type paints: solvent based toluene, xylene), water based, thinners(benzene) | Occupational exposure history questionnaire | 5.73±2.31 | | Painters (both smokers and non-smokers) showed significant reductions (p < 0.05) in PEFR, FVC, FEV1, FEV1/ FVC, and FEF25–75% values. | 5 |
| | Mandal et al., 2013 [12] | India | Paint industry workers/ matched population controls | 149/141 | 43.68 ± 10.63 / 38.07 ± 13.24 | Paint pigments, extenders, binders, solvents & additives: Xylene & other VOCs like, aliphatic hydrocarbons, ethyl acetate,glycolic ethers and acetone | Air sampling in different sections of paint industry | 15.60 ±7.7 | Painters had higher risk of developing chest tightness [1.55 (0.66, 3.64)] & chronic bronchitis [1.43 (0.61, 3.337)] with increase in duration of exposure (>20 years) | Exposure to VOCs increased restrictive lung impairment (79.19%). A significant negative correlation (p<0.001) was seen between PFT and exposure duration (FEV1: r = -0.61, p<0.001). | 5 |

(Continued)

Table 2. (Continued)

| SI No | Author & Year of Publication* | Country | Population (Exposed/Unexposed) | Sample size (Exposed/Unexposed) | Age (Mean ±SD) (Exposed/Unexposed) | Major Exposures & their constituents | Exposure Assessment Methods | Mean Exposure years | Outcome assessed Respiratory Symptoms | Outcome assessed PFT | Quality |
|---|---|---|---|---|---|---|---|---|---|---|---|
| | Hakim et al., 2014 [14] | Egypt | Construction painters/ administrative staffs | 50/50 | 33.6±14.8/ 34.8 ± 15.26 | Paints & organic solvents: benzene | Exposure history questionnaire | 5±2 | Dyspnoea was higher among painters with > 5 years (p-0.008), other symptoms did not show any significant. | No significant difference in pulmonary function | 7 |
| | Aribo et al., 2014 [32] | Nigeria | Automobile painters/ matched population controls | 154/154 | 33.42±0.2/ 32.95±0.70 | Spray paints: solvents (styrene, xylene), pigments like acrylates and methyl acrylates. | Self-structured questionnaire | - | | Spray painting diminishes (p<0.01) lung function (FVC, FEV) but does not significantly affect FEV1% between groups. | 6 |
| | Ojo et al., 2017 [33] | Nigeria | Industrial painters/ electronic technicians | 120/120 | 32.68 ± 13.84 / 33.87±15.48 | Spray paints: organic solvent | Air sampling of Total VOCS (TVOCs) | 12.8 ± 13.7 | | FVC% (p< 0.001), FEV1 (p = 0.002) & FEV1/FVC % (p = 0.005) were significantly lower among spray painters. | 7 |
| | Khode et al., 2017 [34] | India | Automobile painters/ demographically matched unexposed employees | 58/52 | 28.60 ± 5.5 / 25.8 ± 3.8 | Spray paints: solvent fumes | Occupational exposure history questionnaire | 5.9 | Prevalence of dyspnoea (10.3%), nose (20.7%) and throat(8.6%) irritation, and chronic bronchitis (6.89%) were significantly (p<0.0001) higher among painters | Significantly reduced PFT (P < 0.05) among painters. | 6 |
| | Hagras et al., 2017 [35] | Egypt | Industrial painters / employees from same industry | 30/116 | 36.00 ± 2.45 /34.95 ± 2.05 | Spray paint:VOCs (xylene,toluene, isopropanol) | - | 20.73 ± 1.43 | Painters showed lower FEV1, FVC, and FEV1/FVC compared to unexposed individuals (p<0.0001), which was linked to exposure duration. | | 5 |

(Continued)

**Table 2.** (Continued)

| SI No | Author & Year of Publication* | Country | Population (Exposed/Unexposed) | Sample size (Exposed/Unexposed) | Age (Mean ±SD) (Exposed/Unexposed) | Major Exposures & their constituents | Exposure Assessment Methods | Mean Exposure years | Outcome assessed Respiratory Symptoms | PFT | Quality |
|---|---|---|---|---|---|---|---|---|---|---|---|
| | Onesmo et al., 2018 [15] | Tanzania | Construction painters/ matched population controls | 172/148 | 27.9 ± 6.1/ 27.8 ±7.3 | Oil based paints and coatings: VOCs | Not specified | 5.11 ± 4.4 | Dyspnoea, cough, phlegm, nasal irritation, and shortness of breath were higher among painters (p<0.05). Smoking, a lack of knowledge, and inadequate PPE usage were predominant cause of respiratory symptoms. | Painters reported 28% obstructive, 12% restrictive, and 32% mixed airway problems. | 5 |
| | Hwang et al., 2018 [36] | Korea | Industrial painters / employees from same industry | 279/164 | 41.8±7.2/ 39.5± 8.1 | Paints & organic solvents,thinners: xylene, ethylbenzene, toluene,& 2-ethoxyethanol and 2-ethoxyacethyl acetate exposure | Cumulative Exposure Index, Urine analysis for methylhippuric acid, mandelic acid, hippuric acid & 2-Ethoxyacetic acid | 12.7 ± 3.7 | | Organic solvents induces more obstructive than restrictive pulmonary dysfunction. Increased exposure decreased FEV1/FVC and MMF (p<0.01). | 7 |
| | Saraei et al., 2019 [37] | Iran | Automobile painters/ assembly workers | 431/389 | 37.72±2.57/ 36.74±4.36 | Spray paints: VOCs, diisocyanates, | Not specified | 13.01±3.15 | | Painters >10 years had reduced FEV1/FVC (P = 0.005), FEV1 (P = 0.008), and FEF25-75 (P = 0.003). Solvent- and water-based painters exhibited similar spirometry outcomes. | 5 |
| | Jabbar et al., 2020 [38] | Iraq | Construction painters/ matched population controls | 76/25 | 40.93 ± 6.28 / 37.68 ± 6.51 | Paints & organic solvents | Occupational history questionnaire | >5 years | | FEV1, FVC & FEV1% were significantly lower (p<0.0001) among the painters and is highly attributable to exposure duration | 6 |

*(Continued)*

**Table 2.** (Continued)

| SI No | Author & Year of Publication* | Country | Population (Exposed/Unexposed) | Sample size (Exposed/Unexposed) | Age (Mean ±SD) (Exposed/Unexposed) | Major Exposures & their constituents | Exposure Assessment Methods | Mean Exposure years | Outcome assessed | | Quality |
|---|---|---|---|---|---|---|---|---|---|---|---|
| | | | | | | | | | **Respiratory Symptoms** | **PFT** | |
| | Ahmad et al., 2020 [9] | Pakistan | Industrial painters/ administrative staffs | 162/150 | $30.8 \pm 8.4$ / $31.2 \pm 5.8$ | Paints, solvents | Occupational history questionnaire | $7.0 \pm 4.3$ | Exposure increased chest tightness, whistling, asthma-like symptoms, rhinitis, and chronic bronchitis, but not significantly. | FEV1, FEV1/FVC % decreased among exposed but were not statistically significant. | 6 |
| | Ojo et al., 2020 [11] | Nigeria | Industrial painters/ electronic technicians | 120/120 | $32.7 \pm 13.84$ / $33.9 \pm 15.5$ | Spray paints: organic solvent | Air sampling of Total VOCS (TVOCs) | $12.8 \pm 13.7$ | Spray painters had significantly higher rates of recurrent cough ($\chi2 = 11.5$, P = 0.001), breathlessness (LR = 9.9, P = 0.002) and chest pain (LR = 12.8, P<0.001) | | 7 |

**Table 3. Effects of VOC on respiratory symptoms.**

| S. No | Author & year of publication | Sample size (Exposed /Unexposed) (n)* | Cough n (%)$ | Cough COR (95% CI) | Dyspnoea n(%)$ | Dyspnoea COR (95% CI) | Nasal/throat irritation n(%)$ | Nasal/throat irritation COR (95% CI) | Wheezing n(%)$ | Wheezing COR (95% CI) |
|---|---|---|---|---|---|---|---|---|---|---|
| 1. | Schwartz et al.,1988 [10] | 118/299 | 40(34)/71 (23.7) | 1.65 (1.03, 2.62) | 48(40.6)/ 22 (7.4) | 8.63 (4.89, 15.25) | - | - | 40(34)/27 (9) | 5.17 (2.98, 8.95) |
| 2. | Alexandersson et al., 1988 [22] | 38/18 | 2(5) /0 | 2.53 (0.12, 55.55) | 4(10.5)/0 | 4.83 (0.25, 94.62) | 15(39.5)/0 | 24 (1.37, 435.3) | - | - |
| 3. | Hammond et al., 2005 [24] | 106/332 | 43(40.6)/ 106(32) | 1.46 (0.93, 2.28) | - | - | - | - | - | - |
| 4. | Kaukiainen et al., 2005 [25] | 523/505 | 201(38)/ 136(27) | 1.69 (1.3, 2.21) | 72(14)/ 41 (8) | 1.81 (1.21, 2.71) | 237(45)/ 168(33) | 1.66 (1.29, 2.14) | 105(20)/78 (15.6) | 1.35 (0.98,1.86) |
| 5. | Eifan et al., 2005 [23] | 72/72 | 26(36)/13 (18) | 2.57 (1.19, 5.54) | 16(22) / 3 (4) | 6.57 (1.82,23.69) | - | - | 23(32) /12 (16) | 2.35 (1.06, 5.19) |
| 6. | Ould-Kadi et al., 2007 [26] | 106/123 | 9(8.6)/1 (0.8) | 11.56 (1.4, 92.8) | - | - | - | - | - | - |
| 7. | Kaukiainen et al., 2008 [27] | 288/505 | 112(39)/ 125(25) | 1.93 (1.42, 2.64) | 36(12.5)/35 (7) | 1.92 (1.18, 3.13) | 129(45)/ 159(31) | 1.77 (1.31, 2.38) | 55(19) / 73 (14) | 1.4 (0.95, 2.05) |
| 8. | Metwally et al., 2012 [29] | 191/182 | 57(30)/28 (15) | 2.34 (1.41, 3.89) | 56(29)/ 36 (20) | 1.68 (1.04, 2.72) | - | 9. | 0/0 | - |
|  | Khode et al., 2017 [34] | 58/52 | - | - | 6(10)/ 2(4) | 2.88 (0.56, 14.97) | 12(21) / 2 (4) | 6.52 (1.38, 30.71) | - | - |
| 10. | Onesmo et al., 2018 [15] | 172/148 | 91(53)/17 (11.5) | 8.66 (4.81, 15.58) | 114(66)/ 26 (17.6) | 9.22 (5.44, 15.6) | 152(88)/36 (24) | 23.64 (12.99, 43.02) | - | - |
| 11. | Ahmad et al., 2020 [9] | 162/150 | 15(9)/0 | 31.63 (1.88,533.48) | 37(22.7) /15 (10) | 2.66 (1.39, 5.09) | 37 (23) / 15 (10) | 2.66 (1.39, 5.09) | 22(13.5)/ 10 (7) | 2.2 (1.0, 4.82) |
| 12. | Ojo et al., 2020 [11] | 120/120 | 11(5.8)/0 | 25.3 (1.47,434.61) | 7(5.8)/0 | 16 (0.9– 282.04) | - | - | 9(7.5)/ 1 (0.8) | 9.65 (1.2,77.4) |

COR: crude odds ratio, CI: confidence interval, n*: Overall sample size of each study represented as frequency. n(%)$: exposed /unexposed with respiratory symptoms represented as frequency and percentage

*Cough*. Eleven studies that reported cough showed a crude odds ratio which ranged from 1.46 to 31.6. The estimated average crude odds ratio based on the random effects model was û = 2.72 (95% CI: 1.74 to 4.25) and it was statistically significant (p<0.0001) (Fig 2). However, the Q-test indicated heterogeneity (Q = 38, p<0.0001, τ2 = 0.34, I2 = 81.8%). Cook's distance and studentised residual analysis showed that the included studies were neither outliers nor influential.

*Dyspnoea*. Ten studies showed that painters exposed to VOCs had a Crude OR ranging from 1.68 to 15.9. The pooled OR û = 3.59 (95% CI: 2.13 to 6.05) indicated that painters had (p<0.0001) higher odds of dyspnoea compared to those of unexposed individuals (Fig 3) although the Q-test showed heterogeneity (Q = 48.4, p<0.0001, τ2 = 0.46, I2 = 80.23%). However, studentised residual analysis and Cook's distance analysis showed that the included studies did not influence the outcome.

# Cough

Fig 2. Forest plots: Respiratory symptom (Cough) of paint industry workers exposed to VOC.

# Dyspnoea

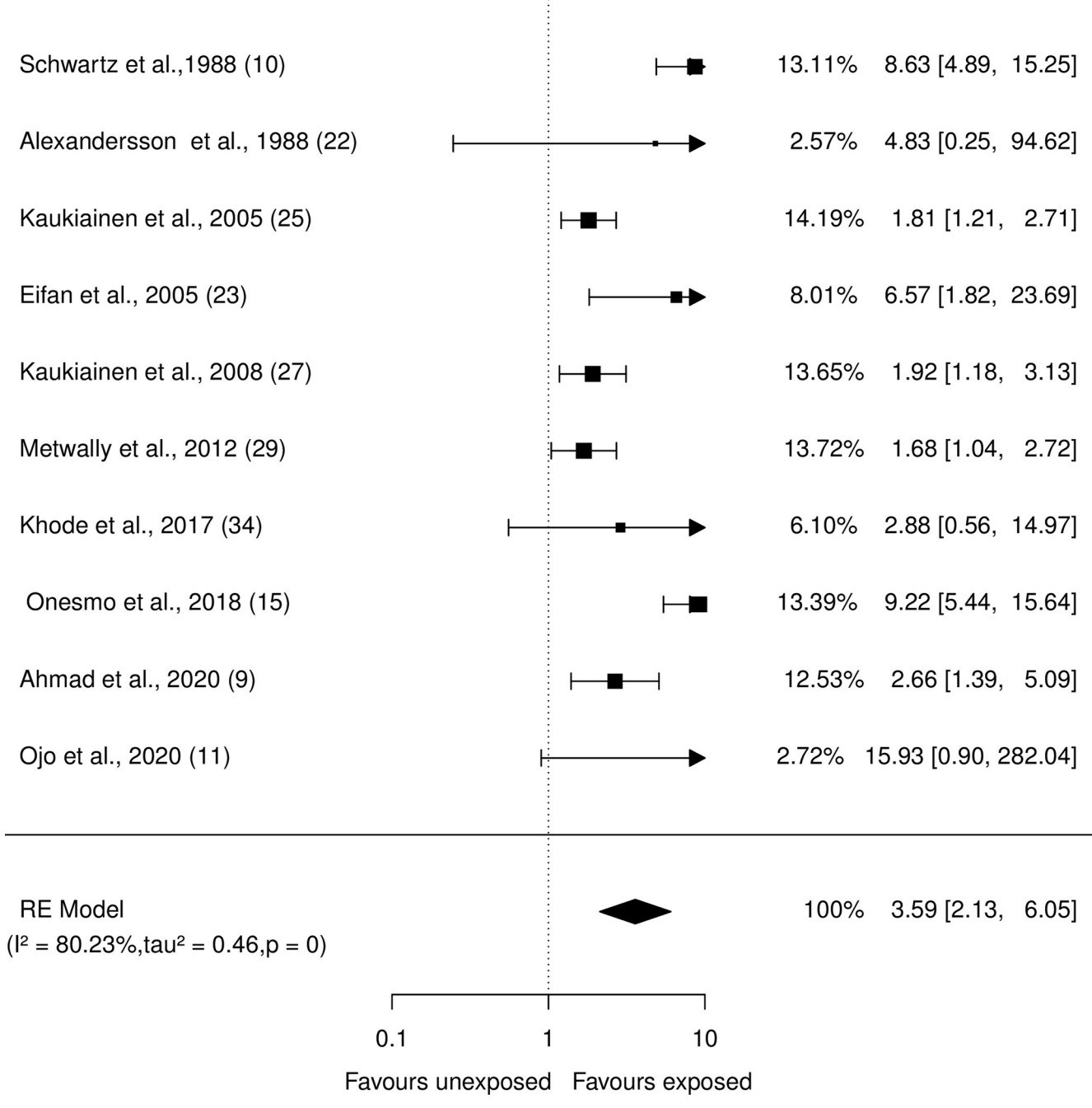

| Study | Weights | Odds Ratio [95% CI] |
|---|---|---|
| Schwartz et al.,1988 (10) | 13.11% | 8.63 [4.89, 15.25] |
| Alexandersson et al., 1988 (22) | 2.57% | 4.83 [0.25, 94.62] |
| Kaukiainen et al., 2005 (25) | 14.19% | 1.81 [1.21, 2.71] |
| Eifan et al., 2005 (23) | 8.01% | 6.57 [1.82, 23.69] |
| Kaukiainen et al., 2008 (27) | 13.65% | 1.92 [1.18, 3.13] |
| Metwally et al., 2012 (29) | 13.72% | 1.68 [1.04, 2.72] |
| Khode et al., 2017 (34) | 6.10% | 2.88 [0.56, 14.97] |
| Onesmo et al., 2018 (15) | 13.39% | 9.22 [5.44, 15.64] |
| Ahmad et al., 2020 (9) | 12.53% | 2.66 [1.39, 5.09] |
| Ojo et al., 2020 (11) | 2.72% | 15.93 [0.90, 282.04] |
| RE Model ($I^2$ = 80.23%,tau$^2$ = 0.46,p = 0) | 100% | 3.59 [2.13, 6.05] |

0.1  1  10

Favours unexposed   Favours exposed

**Fig 3. Forest plots: Respiratory symptom (Dyspnoea) of paint industry workers exposed to VOC.**

# Nasal/Throat irritation

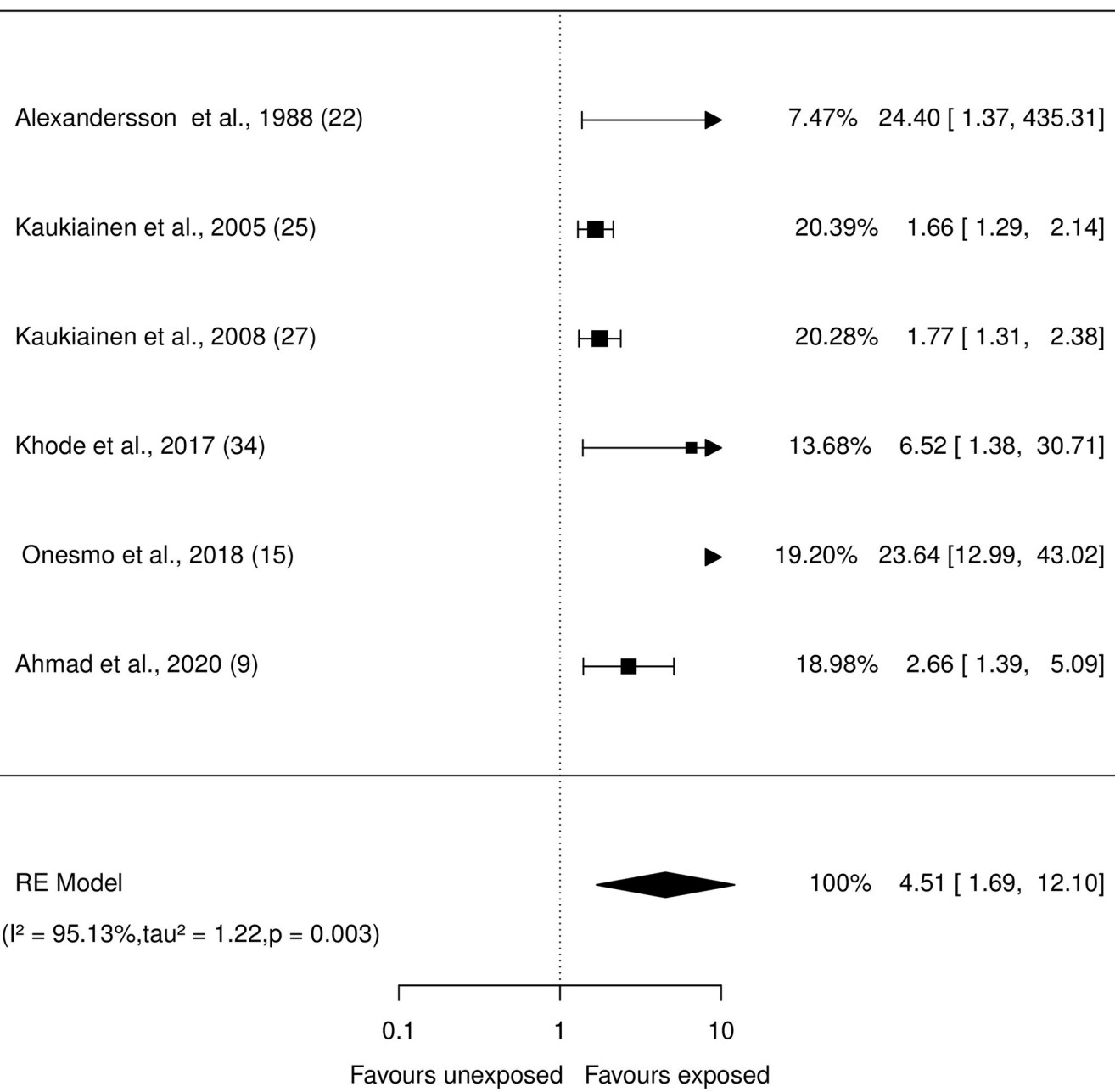

**Fig 4. Forest plots: Respiratory symptom (Nasal/throat irritation) of paint industry workers exposed to VOC.**

# Wheezing

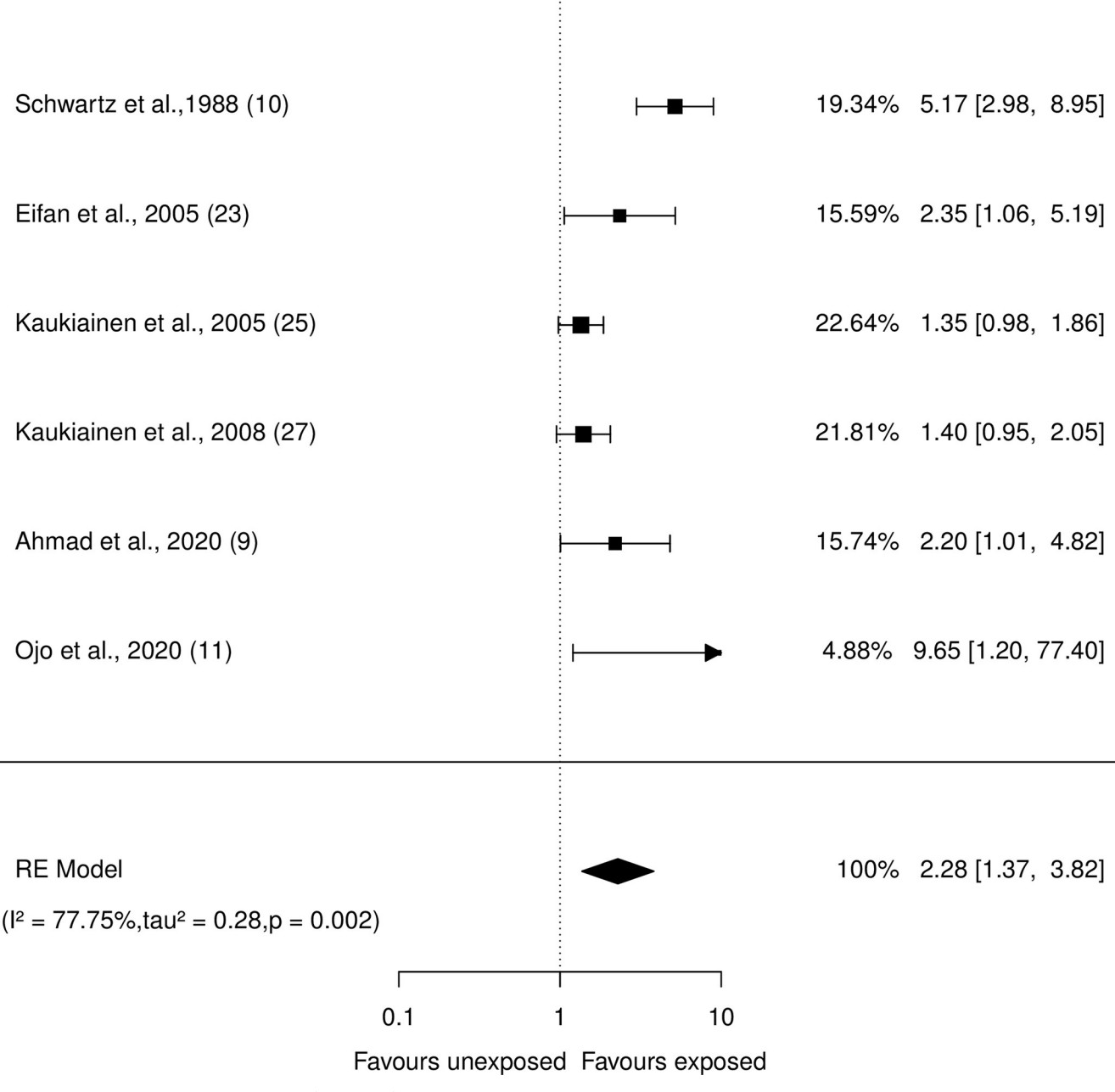

**Fig 5. Forest plots: Respiratory symptom (Wheezing) of paint industry workers exposed to VOC.**

*Nasal/throat irritation*. In connection to nasal/throat irritation, six studies that were chosen for the review revealed that painters exposed to VOCs had an OR ranging from 1.66 to 24.4. The average OR û = 4.5 (95% CI: 1.7 to 12.1) indicated that the painters had increased odds of nasal/throat irritation (p = 0.0027) (Fig 4) although the Q-test showed heterogeneity (Q = 72.04, p<0.0001, $\tau^2$ = 1.22, $I^2$ = 95.13%). Studentised residual analysis revealed that one of the studies [15] could be a potential outlier, while Cook's distance analysis indicated that it was overly influential. Nevertheless, the effect size decreased but remained statistically significant even after the exclusion of the aforementioned study. [(OR û = 1.82 (95% CI: 1.51 to 2.19) p<0.0001);Q = 7.58, p<0.10, $\tau^2$ = 0, $I^2$ = 0%)].

*Wheezing*. A crude OR between 1.35 and 9.65 was reported in six studies for wheezing. The pooled OR was û = 2.28 (95% CI: 1.37 to 3.82), indicating that painters exposed to VOCs had a higher odds of (p = 0.0016) wheezing (Fig 5) than unexposed individuals although the Q-test showed heterogeneity (Q = 22.1, p = 0.0005, $\tau^2$ = 0.28, $I^2$ = 77.75%). Studentized residual tests revealed one of the studies to be a potential outlier [10] and Cook's distance analysis indicated that no studies influenced the outcome. However, after excluding the same, the effect size decreased but remained significant. [(OR û = 1.52 (95% CI: 1.21 to 1.9) p = 0.0003);Q = 5.75, p = 0.22, $\tau^2$ = 0, $I^2$ = 0.01%)].

**Effect of VOCs in paints on PFT parameters.** Among the cross-sectional studies which assessed pulmonary function using spirometry, eighteen out of 19 studies were considered for the meta-analysis. The study by Onesmo et al [15] was excluded from quantitative synthesis as PFT was not assessed among the control group.

The sample size consisted of 2494 participants who had been exposed to VOCs present in paints and 2045 unexposed participants. All studies included male participants, with the mean age of exposed and unexposed participants being 38±9.4 and 37.5±10.9 years respectively. The average duration of exposure to VOCs was 12.6±6.7 years. The mean and standard deviation of the PFT values are summarised in Table 4.

An array of Forest plots depicting the observed outcomes and the estimates based on the random-effects model are shown in Figs 6–8.

*FEV1*. Eighteen studies were included in this quantitative analysis (Fig 6). The observed SMDs ranged from -7.36 to 0.86. The estimated average SMD based on the random-effects model was û = -0.88 (95% CI: -1.5 to -0.2), suggesting that painters exposed to VOCs had a significantly (p = 0.01) lower FEV1 than that of unexposed individuals. The *Q*-test showed heterogeneity (*Q* = 331.56, p<0.0001, $\tau^2$ = 2.05, $I^2$ = 99.05%) across the studies. The studentised residual value revealed one study [30] as a potential outlier. Cook's distance analysis indicated that this study was influential over the outcome. Nevertheless, after excluding this study, the effect size reduced but remained statistically significant [(û = -0.56 (95% CI: -0.85 to -0.28, p<0.0001);Q = 239.8, p<0.0001, $\tau^2$ = 0.32, $I^2$ = 94.58%).

*FVC*. Sixteen studies that evaluated the FVC of painters (Fig 7), showed that their observed SMD ranged from −24.4 to 0. 57. The pooled SMD was û = -2.14 (95% CI: −5.04 to 0.77), (p = 0.14), suggesting a non-significant decrease in the FVC of painters exposed to VOCs. Q-test revealed heterogeneity (Q = 850.5, p<0.0001, $\tau^2$ = 35.13, $I^2$ = 99.94%). Studentised residuals and Cook's distance analysis revealed that one study [32] was a potential outlier and three studies [12, 30, 32] were over influential. The effect size decreased substantially but became statistically significant after these studies were restricted [û = -0.24 (95% CI: -0.44 to -0.05, p = 0.01); Q = 67.92, p<0.0001, $\tau^2$ = 0.93, $I^2$ = 83.47%].

*FEV1/FVC ratio*. Meta-analysis of fourteen studies that assessed the FEV1/FVC ratio (Fig 8) showed SMDs ranging from −4.36 to −0.05. The estimated average SMD was û = -0.97 (95% CI: −1.62to −0.32), suggesting that the painters showed a significant (p = 0.003) reduction in the FEV1/FVC ratio. The Q-test showed heterogeneity (Q = 245.04, p<0.0001, $\tau^2$ = 1.59, $I^2$ =

**Table 4. Effect of VOC on pulmonary function test parameters.**

| S.No | Author & year of publication | Population (Exposed /Unexposed) n | FEVI (Mean ± SD) | | FVC (Mean ± SD) | | FEVI/FVC (Mean ± SD) | |
|---|---|---|---|---|---|---|---|---|
| | | | Exposed | Unexposed | Exposed | Unexposed | Exposed | Unexposed |
| 1. | **Schwartz et al.,1988** [10] | 117/311 | 90.7 ± 16.3* | 94.25 ± 18.9* | - | - | 74.25± 7.6 | 76.6±8.75 |
| | | 117/286 | | | | | | |
| 2. | **Alexandersson et al., 1988** [22] | 38/18 | 4.2 ±0.6** | 4.5 ± 0.55** | 5.35 ± 0.8** | 5.6 ± 0.7** | | |
| 3. | **Eifan et al., 2005** [23] | 62/60 | 101.98±1.9* | 100.4 ± 1.75* | 102.35 ± 2.2* | 101.1 ± 2.2* | 87.2±0.75 | 88.85± 0.7 |
| 4. | **Ould-Kadi et al., 2007** [26] | 106/123 | 96.2 ± 13.4* | 102.7 ± 12.4* | 97.8 ± 12.9* | 103.9 ± 12.3* | 81.8 ± 7.3 | 82.1 ± 6 |
| 5. | **El Mahdy et al., 2009** [28] | 38/30 | 91.6±1.9* | 93±1.32* | 93.7±2.14* | 95.13±1.38 | 80.84±0.8 | 83.3±0.9 |
| 6. | **Metwally et al., 2012** [29] | 191/182 | 80.9±15.9* | 91.97±21.5* | 70.9±18.15* | 81.5 ± 9.7* | - | - |
| 7. | **Numan 2012** [30] | 30/30 | 50.4 ± 5.2* | 81.2±2.7* | 70 ±4.34* | 88.6±2.2* | 72.8±6.8 | 95 ±1.9 |
| 8. | **El-Gharabawy et al., 2013**[31] | 30/30 | 71.23±4.95* | 79.1±5.2* | - | - | 90.25±6.3* | 95.7±6.7* |
| 9. | **Mandal et al., 2013** [12] | 149/141 | 2.65± 0.5** | 3.5 ± 0.6** | 2.9 ± 0.57** | 3.95± 0.6** | | |
| 10. | **Hakim et al., 2014** [14] | 36/36 | 3.8 ± 0.5** | 4.02 ± 0.4** | 4.5 ± 0.6** | 4.75 ± 0.5** | 84.5 ±4.45 | 84.5± 3.7 |
| 11. | **Aribo et al., 2014** [32] | 154/154 | 2.7±0.1** | 3.35±0.7** | 2.7 ± 0.1** | 4.4±0.1** | - | - |
| 12. | **Ojo et al., 2017** [33] | 120/120 | 3.1±0.6 ** | 3.3±0.5** | 3.6±0.6** | 3.7±0.5** | 85.5±8.7 | 87.9 ±6.2 |
| 13. | **Khode et al., 2017** [34] | 58/52 | 3.04 ± 0.7** | 3.2 ± 0.4** | 3.3 ± 0.7** | 3.4 ± 0.37** | 86.9 ± 7.7 | 91.5 ± 6.2 |
| 14. | **Hagras et al., 2017** [35] | 116/30 | 79.04 ±67.4* | 93.65 ±39.9* | 83.8±62.1* | 94.7±36.4* | 95.6±49.5* | 98.1±35.8* |
| 15. | **Hwang et al., 2018** [36] | 580/164 | 3.45 ± 0.6** | 3.5 ± 0.7** | 4.2 ± 0.65** | 4.1 ± 0.8** | 81.8± 5.43 | 85.6± 6.7 |
| 16. | **Saraei et al., 2019** [37] | 431/389 | 92 ± 11* | 94 ± 15* | 94 ± 10* | 95 ± 17* | 78.6±5.5 | 79.6±4.7 |
| 17. | **Jabbar et al., 2020** [38] | 76/25 | 2.8 ±0.6** | 3.5 ± 0.4** | 3.78 ± 0.5** | 4.03 ±0.4** | 73.4±10.6 | 87.04±4.6 |
| 18. | **Ahmad et al., 2020** [9] | 162/150 | 3.12 ± 0.5** | 3.5 ± 0.4** | 4.12 ± 0.6** | 4.4 ± 0.6** | 79.6 ± 14.3 | 80.8 ± 6.6 |

FEV1: Forced expiratory volume in the first second, FVC: Forced vital capacity

* values represented as % predicted

** values represented as litres

98.61%). Studentised residuals and Cook's distance analysis revealed one study [30] as a potential outlier that could have excessively influenced the outcome. Despite removing the study, the effect size was still significant [(û = -0.72 (95% CI: -1.2 to -0.24, p = 0.003); Q = 175.34, p<0.0001, $\tau^2$ = 0.73, $I^2$ = 97.4%].

**Publication bias.** Visual inspection of the funnel plot and regression test of funnel plot asymmetry for respiratory symptoms (cough: p = 0.06, dyspnoea: p = 0.33, nasal/throat irritation: p = 0.18, wheezing: p = 0.1) showed no formal evidence of publication bias (Fig 9). Moreover, the robustness of the current meta-analysis was proved by a file-drawer analysis (failsafe N: 226 >65, N: 217>60, N: 125>40, for cough, dyspnoea, and nasal/throat irritation respectively).The results showed that more studies with effect size zero would be required to negate the present findings. Furthermore, application of file drawer analysis for wheezing (failsafe N: 43>40) showed that almost the same number of studies with null values would be required to negate the present findings. Additionally, visual inspection of the funnel plot (Fig 9) and regression test for pulmonary function showed significant asymmetry (p<0.0001), however the file drawer analysis (failsafe N: 1095>100, and N: 423>90, for FEV1 and FEV1/FVC) showed that more studies with null values would be required to negate the present findings, suggesting minimal or no publication bias.

**Subgroup analysis.** Subgroup analysis was conducted in studies that evaluated respiratory symptoms and pulmonary function in each subgroup (geographical area, occupational sector, and smoking status).

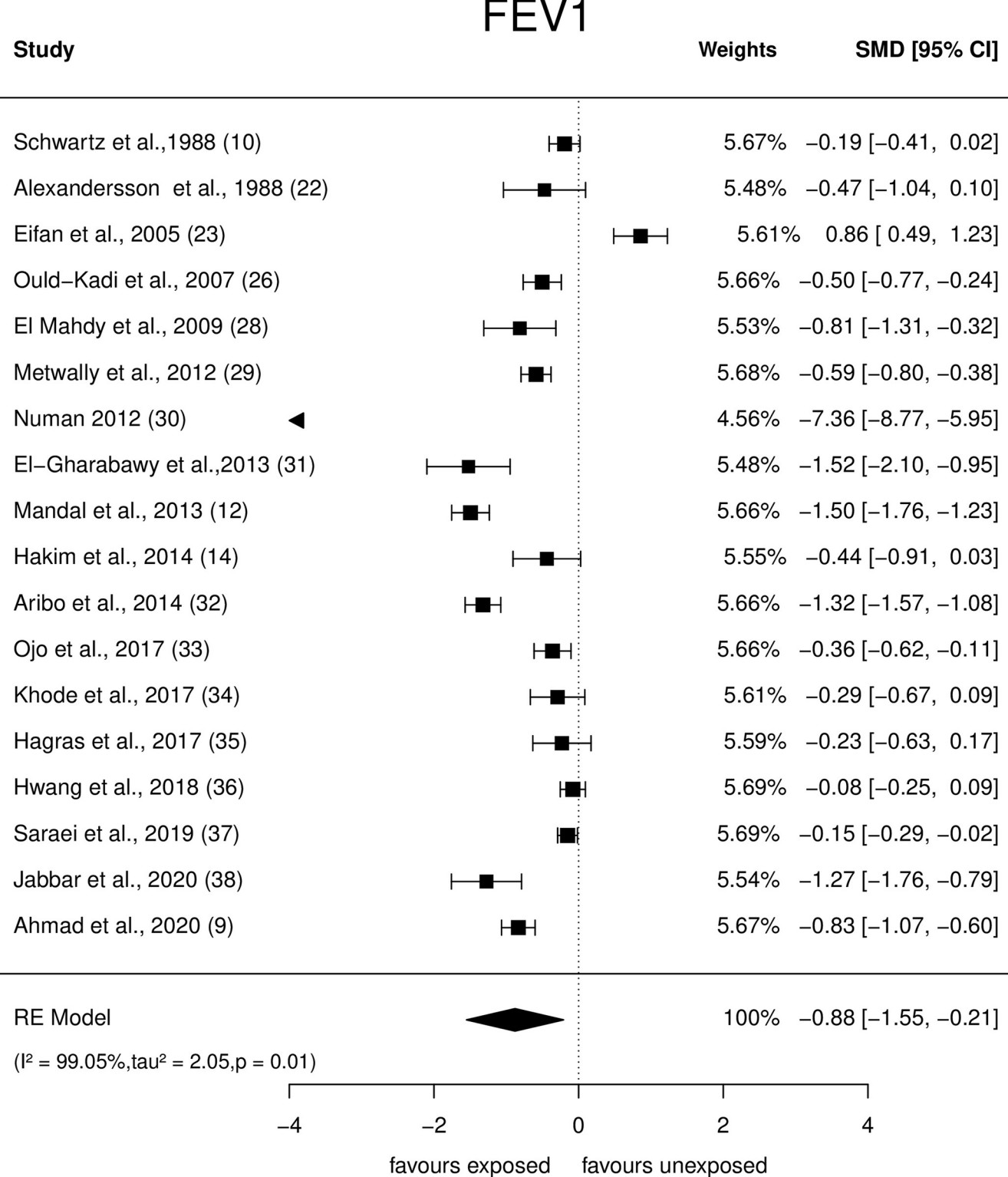

**Fig 6. Forest plots: PFT parameter (FEV1) of paint industry workers exposed to VOC.**

# FVC

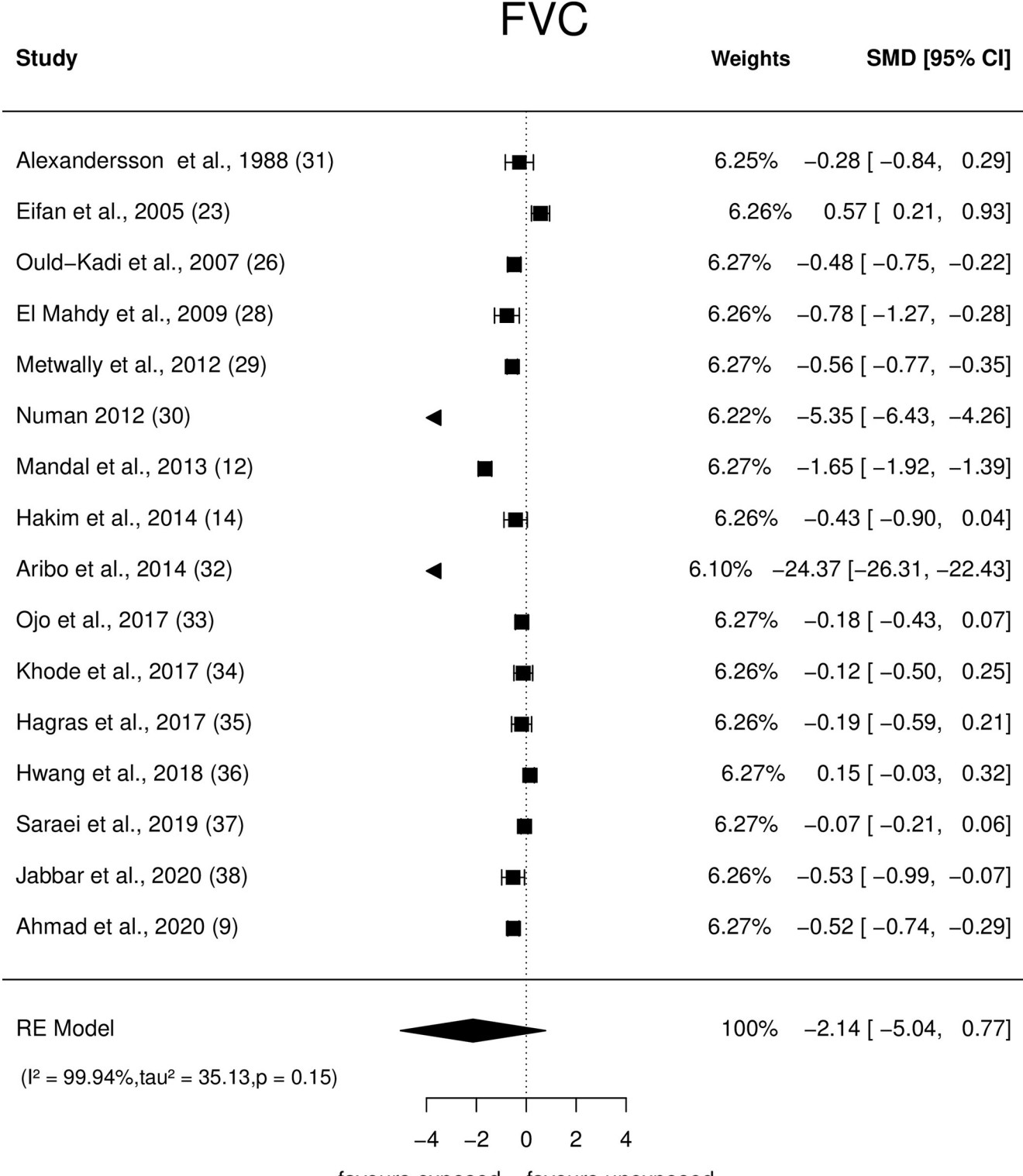

| Study | Weights | SMD [95% CI] |
|---|---|---|
| Alexandersson et al., 1988 (31) | 6.25% | −0.28 [ −0.84, 0.29] |
| Eifan et al., 2005 (23) | 6.26% | 0.57 [ 0.21, 0.93] |
| Ould–Kadi et al., 2007 (26) | 6.27% | −0.48 [ −0.75, −0.22] |
| El Mahdy et al., 2009 (28) | 6.26% | −0.78 [ −1.27, −0.28] |
| Metwally et al., 2012 (29) | 6.27% | −0.56 [ −0.77, −0.35] |
| Numan 2012 (30) | 6.22% | −5.35 [ −6.43, −4.26] |
| Mandal et al., 2013 (12) | 6.27% | −1.65 [ −1.92, −1.39] |
| Hakim et al., 2014 (14) | 6.26% | −0.43 [ −0.90, 0.04] |
| Aribo et al., 2014 (32) | 6.10% | −24.37 [−26.31, −22.43] |
| Ojo et al., 2017 (33) | 6.27% | −0.18 [ −0.43, 0.07] |
| Khode et al., 2017 (34) | 6.26% | −0.12 [ −0.50, 0.25] |
| Hagras et al., 2017 (35) | 6.26% | −0.19 [ −0.59, 0.21] |
| Hwang et al., 2018 (36) | 6.27% | 0.15 [ −0.03, 0.32] |
| Saraei et al., 2019 (37) | 6.27% | −0.07 [ −0.21, 0.06] |
| Jabbar et al., 2020 (38) | 6.26% | −0.53 [ −0.99, −0.07] |
| Ahmad et al., 2020 (9) | 6.27% | −0.52 [ −0.74, −0.29] |
| RE Model | 100% | −2.14 [ −5.04, 0.77] |

($I^2$ = 99.94%, tau² = 35.13, p = 0.15)

−4  −2  0  2  4

favours exposed    favours unexposed

**Fig 7. Forest plots: PFT parameter (FVC) of paint industry workers exposed to VOC.**

# FEV1/FVC

**Fig 8. Forest plots: PFT parameter (FEV1/FVC) of paint industry workers exposed to VOC.**

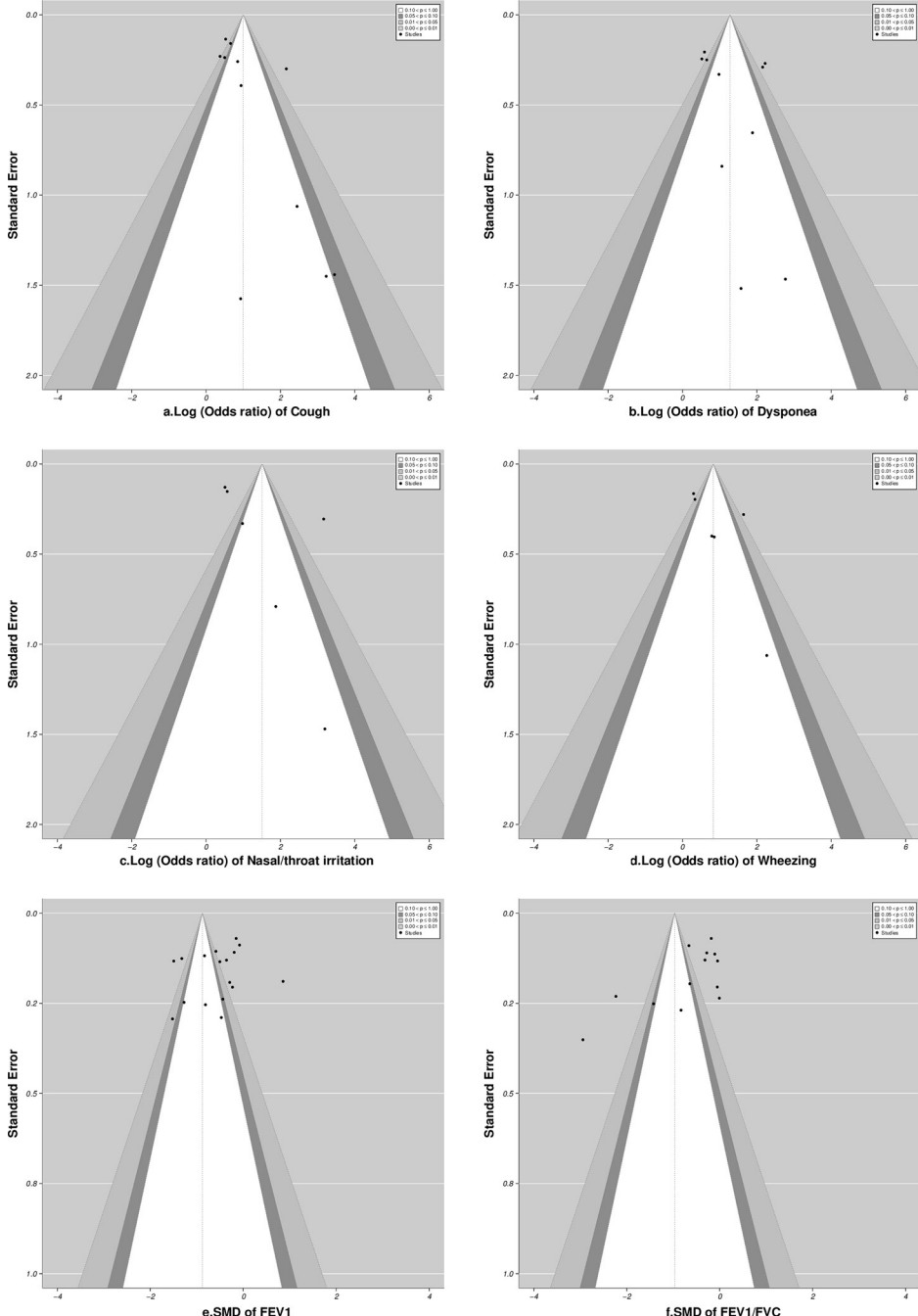

**Fig 9.** Contour−Enhanced Funnel plot: a-d Respiratory symptoms & e & f PFT parameters.

*Subgroup analysis of respiratory symptoms.* Considering geographical areas, the effect size of respiratory symptoms such as cough (OR: 4.71, 95% CI: 2.18 to 10.2), dyspnoea (OR: 5.05, 95% CI: 1.8 to 14.01), and wheezing (OR: 3.38, 95% CI: 1.01 to 11.4) were significantly higher among studies from Africa compared to the overall effect size. This difference was also observed when comparing Africa to Europe and America (Table 5). Additionally, relatively few studies have assessed respiratory symptoms among painters in Asia. Regarding occupational sectors, studies conducted among industrial sectors yielded higher effect sizes for cough

**Table 5. Subgroup analysis of respiratory symptoms based on regions & occupational sectors.**

| Subgroup Variables | | Cough Studies (K) | Cough OR (95% CI) | Cough $I^2$ (%) | Dyspnea Studies (K) | Dyspnea OR (95% CI) | Dyspnea $I^2$ (%) | Nasal/throat irritation Studies (K) | Nasal/throat irritation OR (95% CI) | Nasal/throat irritation $I^2$ (%) | Wheezing Studies (K) | Wheezing OR (95% CI) | Wheezing $I^2$ (%) |
|---|---|---|---|---|---|---|---|---|---|---|---|---|---|
| **All Studies** | | 11 | 2.72 (1.74,4.25) | 81.8 | 10 | 3.59 (2.13,6.05) | 80.2 | 6 | 4.51 (1.69,12.1) | 95.1 | 6 | 2.29 (1.37,3.82) | 77.75 |
| **Region** | Europe & USA | 5 | 1.72 (1.45,2.03) | 0 | 4 | 3.16 (1.28,7.69) | 87.2 | 3 | 1.73 (1.42,2.09) | 0 | 3 | 2.07 (0.89,4.85) | 92.2 |
| | Africa | 5 | 4.71 (2.18,10.2) | 71.7 | 4 | 5.05 (1.8,14.01) | 82.6 | 1 | 23.57 (12.93,42.94) | - | 2 | 3.38 (1.01,11.4) | 35.33 |
| | Asia | 1 | 31.5 (1.9,533.8) | - | 2 | 2.69 (1.47,4.9) | 0 | 2 | 3.13 (1.6,6.11) | 8.4 | 1 | 2.2 (1.01,4.8) | - |
| **Occupational sector** | Construction sector | 4 | 2.53 (1.2,5.36) | 93.6 | 4 | 4.01 (1.65,9.8) | 92.3 | 3 | 4.01 (0.74,21.97) | 98.6 | 3 | 2.07 (0.9,4.85) | 92.2 |
| | Industrial sector | 5 | 3.35 (1.18,9.4) | 66.8 | 5 | 3.45 (1.97,6.05) | 3.3 | 3 | 4.26 (1.6,11.36) | 33.9 | 3 | 2.51 (1.46,4.3) | 0 |
| | Paint manufacturing sector | 2 | 3.74 (0.89,15.5) | 53.1 | 1 | 1.68 (1.04,2.72) | - | 0 | - | - | 0 | - | - |

OR: crude odds ratio, CI: confidence interval, $I^2$ –level of heterogeneity, K: number of studies

(OR: 3.35, 95% CI: 1.18 to 9.4) and wheezing (OR: 2.51, 95% CI: 1.46 to 4.3) in comparison to the overall effect size. Moreover, in comparison to the overall effect size, symptoms such as dyspnoea (OR: 3.45, 95% CI: 1.97 to 6.05) were more prevalent in studies done among painters working in construction sector. Furthermore, there is a scarcity of research that has examined respiratory symptoms in paint manufacturing sectors (Table 5).

*Subgroup analysis of pulmonary function.* On performing a subgroup analysis for pulmonary function (Table 6), the SMD of FEV1/FVC (SMD: -1.3, 95% CI: -2.4 to -0.25) were

**Table 6. Subgroup analysis of pulmonary function test based on regions and occupational sectors.**

| Subgroup Variables | | FEV1 Studies (K) | FEV1 SMD (95% CI) | FEV1 $I^2$ (%) | FVC Studies (K) | FVC SMD (95% CI) | FVC $I^2$ (%) | FEV1/FVC Studies (K) | FEV1/FVC SMD (95% CI) | FEV1/FVC $I^2$ (%) |
|---|---|---|---|---|---|---|---|---|---|---|
| **All studies** | | 18 | -0.88 (-1.55,-0.21) | 99.06 | 16 | -2.14 (-5.04,0.77) | 99.59 | 14 | -0.97 (-1.62,-0.32) | 98.62 |
| **Region** | Europe & USA | 2 | -0.23 (-0.43,-0.03) | 0 | 1 | -0.28 (-0.83,0.28) | - | 1 | -0.28 (-0.49,-0.06) | - |
| | Africa | 8 | -0.70 (-1.01,-0.4) | 86.5 | 7 | -3.8 (-10.5,2.8) | 99.96 | 6 | -0.7 (-1.5,0.2) | 96.9 |
| | Asia | 8 | -1.3 (-2.9,0.4) | 99.7 | 8 | -0.9 (-2.1,0.35) | 99.47 | 7 | -1.3 (-2.4,-0.25) | 99.13 |
| **Occupational sector** | Construction sector | 4 | -0.83 (-1.46,-0.2) | 89.55 | 2 | -0.48 (-0.8,-0.16) | 0 | 4 | -0.62 (-1.22,-0.01) | 89.07 |
| | Industrial sector | 11 | -0.94 (-2.12,0.24) | 99.51 | 11 | -2.79 (-7.06,1.47) | 99.96 | 9 | -1.24 (-2.2,-0.27) | 99.13 |
| | Paint manufacturing sector | 3 | -0.86 (-1.48,-0.24) | 94.9 | 3 | -0.9 (-1.64,-0.16) | 96.36 | 1 | -0.05 (-0.3,0.21) | - |

FEV1: Forced expiratory volume in the first second, FVC: Forced vital capacity, SMD: standardised mean difference, $I^2$: level of heterogeneity, K: number of studies

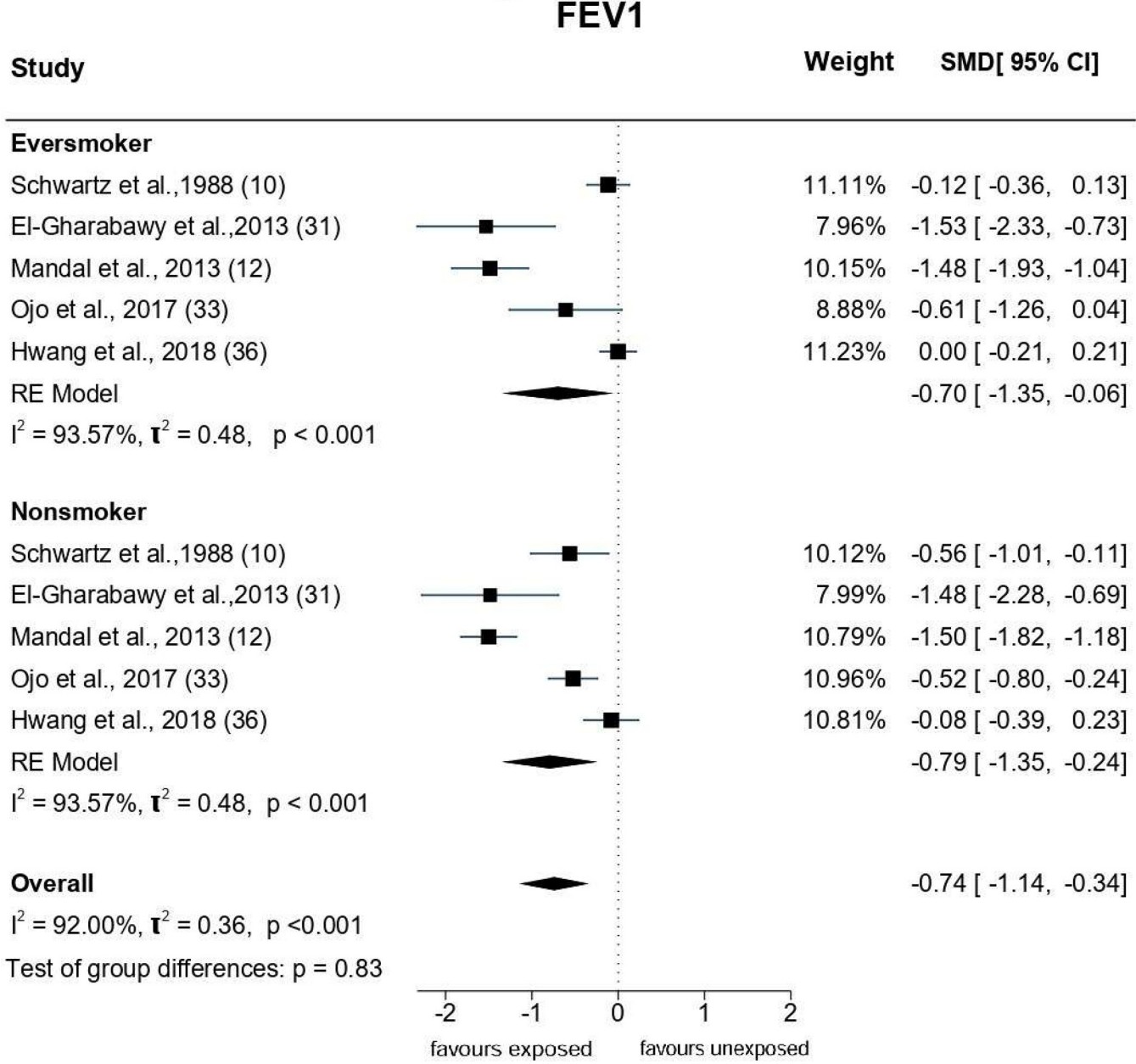

**Fig 10. Subgroup analysis of FEV1 based on smoking status.**

significantly higher in studies conducted in Asia compared to the overall effect size. Moreover, very few studies have assessed pulmonary function among painters in Europe and America. Further, subgroup analysis based on occupational sectors revealed that the effect size of FEV1/FVC (SMD: -1.24, 95% CI: -2.2 to -0.3) was found to be higher in industrial sectors compared to the overall effect size. Further, there is scarcity of research among construction sectors and paint manufacturing sectors (Table 6).

*Subgroup analysis by smoking status.* Figs 10–12 depict the effect size of the PFT parameters in studies that stratified pulmonary function based on smoking status. The results indicated that the estimated average SMD among exposed smokers for FEV1 is û = -0.7(-1.35,-0.06) and

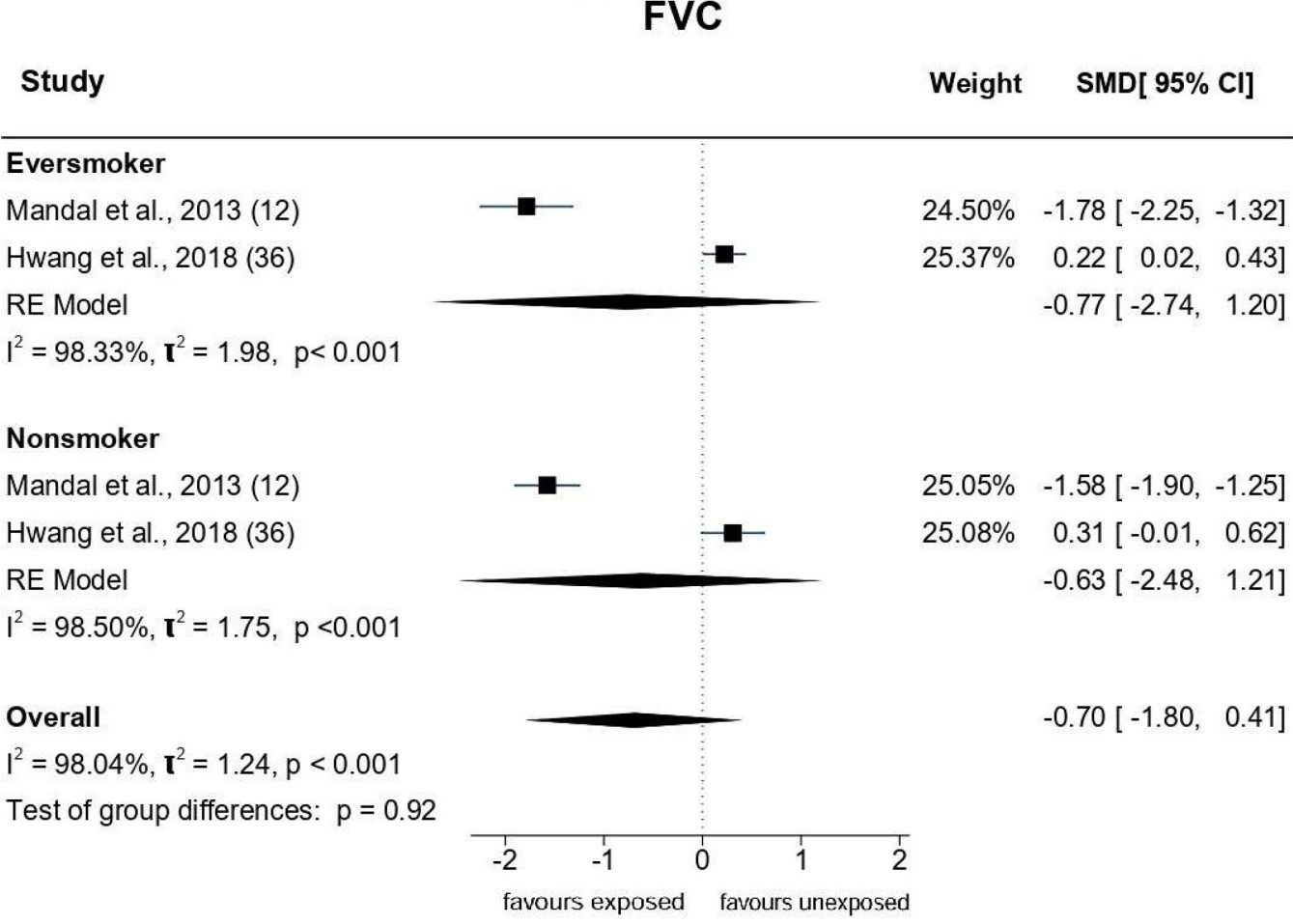

**Fig 11. Subgroup analysis of FVC based on smoking status.**

FEV1/FVC is û = -0.35(-0.53,-0.18), implying that smokers exposed to VOC showed lower pulmonary function compared to smokers who were not exposed to VOC. Furthermore, this tendency is consistent among exposed non-smokers [(FEV1: -0.79(-1.35,-0.24) FEV1/FVC: -0.54(-0.76,-0.33)], demonstrating that VOC exposure reduces pulmonary function regardless of smoking status. However, the subgroup analysis of FVC based on the smoking status did not yield significant difference.

## Discussion

The current systematic review and meta-analysis included twelve observational studies that evaluated respiratory symptoms such as cough, dyspnoea, nasal/throat irritation, and wheezing, and eighteen observational studies that assessed pulmonary function parameters including FEV1, FVC, and FEV1/FVC, using spirometry in paint industry workers. The literature search yielded no meta-analysis on the pulmonary function among paint industry workers in diverse occupational sectors who are exposed to VOCs present in paints and organic solvents. However, other systematic reviews and meta-analysis on the pulmonary health impacts of VOC among the general population revealed minimal or medium-sized effects on pulmonary function, such as on the onset of asthma and wheezing [39].

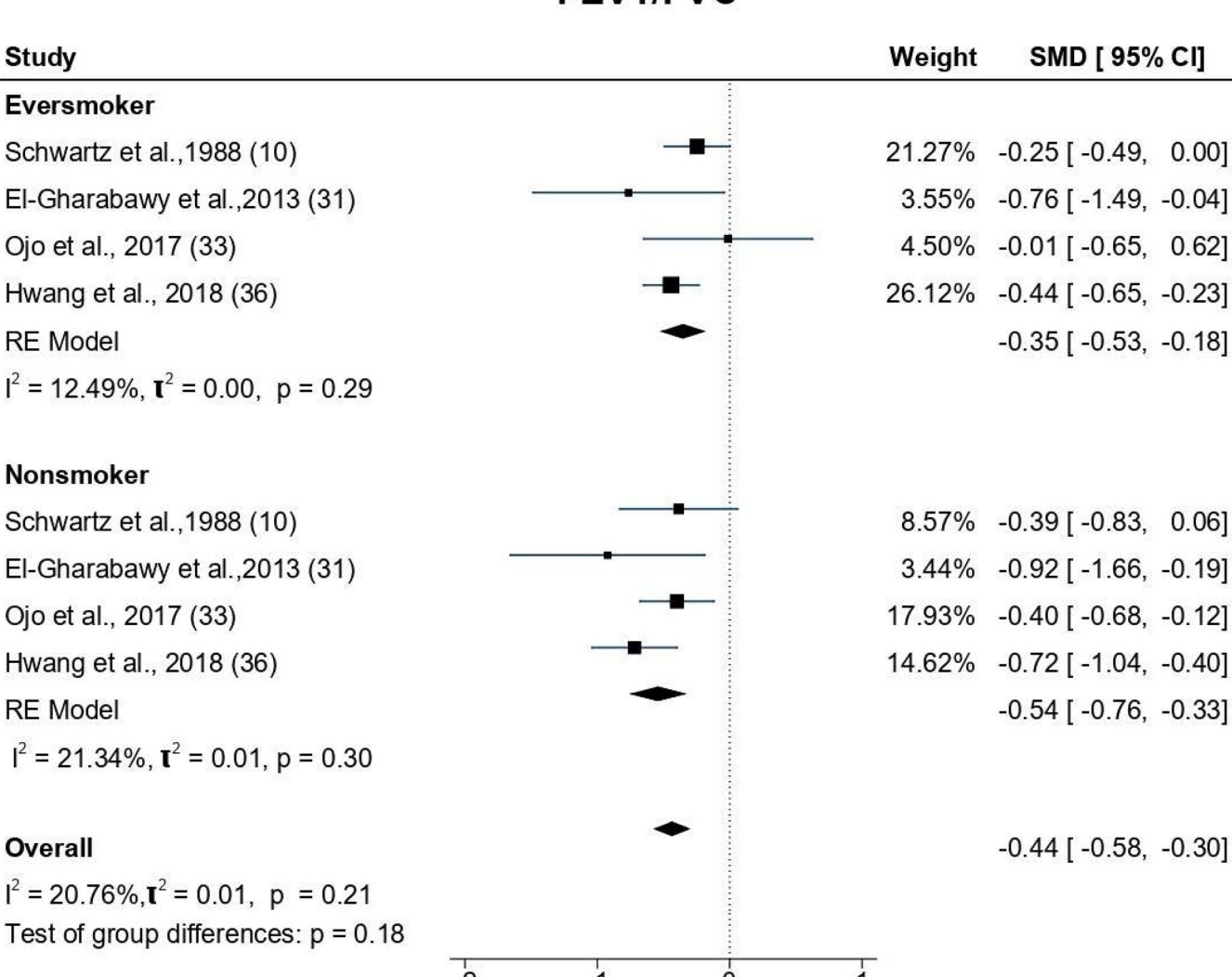

**Fig 12. Subgroup analysis of FEV1/FVC based on smoking status.**

The current meta-analysis of respiratory symptoms (n = 12) showed an increased odds of respiratory symptoms such as cough, dyspnoea, nasal/throat irritation, and wheezing among workers exposed to VOC compared to controls FEV1, and FEV1/FVC were found to be significantly reduced among exposed painters, [10, 23]. Experimental evidence reported in studies indicates that VOCs are lipophilic, penetrate the cell membrane and cytoplasm with ease, and lead to an oxidative stress-induced inflammatory response in the lung parenchyma, resulting in narrowing of the airway and subsequent obstructive airway disease [40].

In the current meta-analysis, it was found that the magnitudes of effect sizes varied based on the geographical region. The effect size of respiratory symptoms was larger in studies conducted in Africa, while studies in Asian nations showed greater reduction in pulmonary function compared to developed countries in Europe and America, as well as the overall effect size. A few of the contributing factor could be a lack of adequate health awareness and knowledge

about the consequences of exposure among employees, as well as an absence of work enforcement policies by employers. Furthermore, poor utilisation and ineffective use of personal protective equipment (PPE) such as masks, respirators, gloves, coveralls, and eye goggles play an additive role and increases the risk of developing respiratory dysfunctions [41, 42]. In addition, owing to the increased ambient warmth and humidity in these places, painters often complain that using PPEs are uncomfortable [15]. Improved air quality, wide use of PPEs by painters in developed nations, and strong work enforcement rules are some of the causes for the decreased respiratory health effect among paint industry workers in developed nations. Furthermore, it was shown that heterogeneity persisted ($I^2$: 71.7%) even after stratifying by geographical region since few of these studies were overly influential on the total effect size and may be considered outliers [15, 30, 32]. However, the effect size remained significant even after omitting these studies.

Comparison of occupational sectors showed that, painters working in industrial sectors had a higher odds of respiratory symptoms such as cough and wheezing, whereas painters working in construction sectors had a higher odds of dyspnoea. Furthermore, painters from industrial sectors had lower FEV1/FVC, when compared to the overall effect size. A probable explanation is the widespread usage of spray painting techniques in the studied occupational areas [11]. Also, there is a scarcity of research that has examined the pulmonary function of workers in unorganised sectors, including minor paint manufacturing plants and the construction industry, where conducting research among these populations might present logistical challenges. It is worth noting that painters employed in these industries might be highly exposed to solvents found in old building paints, which in turn might contain hazardous substances such as benzene and toluene [43]. Furthermore, the painters might be routinely engaged in a range of tasks including paint scraping, cleaning, dissolving raw materials, agitating solvents, and filtration [44].

The current meta-analysis restricted the research population to those aged 15 to 60, as aging might cause anatomical and functional deficits in respiratory functioning [45, 46]. Furthermore, previous meta-analyses of the general population found that exposure to VOCs increased the risk of respiratory dysfunctions in the elderly (>60) [39]. Also, subgroup analysis, which included potential confounders such as smoking, revealed that painters exposed to VOCs had reduced pulmonary function regardless of smoking status. Additionally, studies that used stratification by duration of exposure found that pulmonary function deteriorated with increasing years of exposure [10, 12, 14, 28, 29, 38]. However, the stratification differed between studies, making data aggregation more challenging. Furthermore other factors that might influence the degree of respiratory consequences, such as paint type, concentration in the air, rate of volatilization, and ventilation [8, 24, 33, 47], were not assessed in the current meta-analysis. Most included studies focused on male participants, except for two studies that examined respiratory symptoms in both genders [15, 24]. As a result, the potential influence of gender as a confounding factor could not be determined.

The studies included in this meta-analysis exhibited heterogeneity, which could be attributed to variations in study characteristics such as income level, race, environmental pollutants, climatic changes, occupational types, ventilation status, and awareness of personal protective equipment (PPE) usage. Additionally, some studies had small sample sizes and lacked precision. However, it is worth noting that the statistical power of this analysis is significant. Although publication bias existed among studies that assessed the pulmonary function parameters, Fail-safe (sensitivity) analysis showed that a substantially greater number of studies with null values would be required to negate the present findings. Furthermore, several studies from developing countries, which generally tend to be underrepresented in the occupational

epidemiology literature. The above inclusion has added value and can be considered as one of the major strengths of the current analysis.

However, the present review has certain limitations as it ended up including only cross-sectional studies. Few longitudinal [48] and case-control studies that were identified were excluded for meta-analysis as they lacked control (unexposed) group [48] or they did not quantify pulmonary function [49]. Thus, although the study revealed excess respiratory symptoms and lower mean pulmonary function, the overall quality of evidence might be limited due to the inability to examine temporality and chronic effects in a cross-sectional design. Further, crude odds ratios were calculated to assess effect size of respiratory symptoms, which may not account for key confounders such as age, smoking, gender etc., reducing robustness of the study. In addition, restricting the search strategy to English might lead to linguistic bias and reduced evidence analysis. For inclusivity and generalizability, future assessments should incorporate multilingual research. Subgroup analysis of respiratory symptoms based on smoking status was not done due to the absence of such information in the included studies. Additionally, smokers had been excluded from most studies [14, 30, 34, 35, 37] and few have simply reported smoking habits under descriptive characteristics [9, 12, 22, 23, 26, 28, 29].

It can be inferred from the current review that paint industry workers exposed to VOCs exhibit adverse respiratory health impacts—such as a decline in pulmonary function. Furthermore, this review served as a needs assessment, allowing us to identify gaps in the current knowledge of pulmonary function among painters. Limited research assessed the pulmonary function in developed countries and respiratory symptoms in Asian countries. There is a dearth of sufficient evidence about the influence of VOCs on painters working in unorganized sectors such as construction and paint manufacturing, as compared to organized sectors such as the automobile industry. The prevalence of respiratory ailments, including COPD and pulmonary fibrosis, among painters is not well-documented. However, other meta-analyses have reported that the pooled risk ratio of studies that evaluated the impact of VOC exposure on asthma was higher, particularly for exposures to benzene, toluene, and p-dichlorobenzene [50]. A detailed assessment is needed to assess the impacts of solvent-based paints against water-based paints, as well as the differences between spray painting and brushing/rolling methods. Therefore, it is crucial to do further study in this topic to ascertain the actual impact.

## Conclusion

The current meta-analysis, performed based on a comprehensive literature review, identified a higher prevalence of respiratory symptoms and a lower mean pulmonary function among painters exposed to VOCs compared to the controls. Notably, respiratory symptoms such as cough, dyspnoea, nasal/throat irritation, and wheezing were consistently reported, alongside significant reductions in FEV1 and FEV1/FVC ratios. However, the study's reliance on cross-sectional data limits the ability to establish causality or temporality. Furthermore, the use of crude odds ratios for respiratory symptoms without adjustment for potential confounders like age, smoking, and gender underscores the need for cautious interpretation of the findings.

The meta-analysis revealed significant variations in the strength of associations across geographical regions and occupational sectors, highlighting the complex interplay of environmental factors and workplace practices. These variations underscore the importance of tailored occupational health interventions and policies. Future research should prioritize longitudinal studies, particularly in developing nations and unorganized industrial sectors such as construction sites and small paint manufacturing industries. Specific attention should be given to exploring the impact of different types of VOCs, paint formulations (solvent-based vs. water-based), and painting techniques on respiratory health outcomes. Addressing these gaps will

provide a more nuanced understanding of VOC exposure effects and inform targeted preventive measures in occupational settings.

## Supporting information

**S1 Dataset.**
(XLSX)

**S1 Table. PRISMA 2020 checklist.**
(DOCX)

**S2 Table. Respiratory symptoms and pulmonary function in paint industry workers: A systematic review and meta-analysis.**
(DOCX)

## Acknowledgments

We extend our sincere thanks to the management of SRIHER (DU), Chennai.

## Author Contributions

**Conceptualization:** Lavanya Sekhar, Vidhya Venugopal, Priscilla Johnson.

**Data curation:** Lavanya Sekhar, Akila Govindarajan Venguidesvarane.

**Formal analysis:** Lavanya Sekhar, Gayathri Thiruvengadam, Yogita Sharma.

**Funding acquisition:** Lavanya Sekhar.

**Investigation:** Lavanya Sekhar, Gayathri Thiruvengadam, Yogita Sharma.

**Methodology:** Lavanya Sekhar, Akila Govindarajan Venguidesvarane, Gayathri Thiruvengadam, Yogita Sharma, Priscilla Johnson.

**Resources:** Akila Govindarajan Venguidesvarane, Yogita Sharma.

**Supervision:** Vidhya Venugopal, Santhanam Rengarajan, Priscilla Johnson.

**Validation:** Yogita Sharma, Priscilla Johnson.

**Visualization:** Gayathri Thiruvengadam, Santhanam Rengarajan, Priscilla Johnson.

**Writing – original draft:** Lavanya Sekhar.

**Writing – review & editing:** Lavanya Sekhar, Akila Govindarajan Venguidesvarane, Gayathri Thiruvengadam, Yogita Sharma, Vidhya Venugopal, Priscilla Johnson.

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
