## [Decision Letter · Decision Letter 0]

8 May 2024

PONE-D-24-08732Respiratory symptoms and Pulmonary function in paint industry workers: A Systematic Review and Meta-AnalysisPLOS ONE

Dear Dr. Johnson,

Thank you for submitting your manuscript to PLOS ONE. After careful consideration, we feel that it has merit but does not fully meet PLOS ONE’s publication criteria as it currently stands. Therefore, we invite you to submit a revised version of the manuscript that addresses the points raised during the review process.

We look forward to receiving your revised manuscript.

Kind regards,

Giulia Squillacioti

Academic Editor

PLOS ONE

Journal Requirements:

Reviewers' comments:

Reviewer's Responses to Questions

**Comments to the Author**

1. Is the manuscript technically sound, and do the data support the conclusions?

Reviewer #1: Partly

Reviewer #2: Yes

Reviewer #3: Yes

2. Has the statistical analysis been performed appropriately and rigorously? 

Reviewer #1: No

Reviewer #2: Yes

Reviewer #3: Yes

3. Have the authors made all data underlying the findings in their manuscript fully available?

Reviewer #1: Yes

Reviewer #2: Yes

Reviewer #3: Yes

4. Is the manuscript presented in an intelligible fashion and written in standard English?

Reviewer #1: Yes

Reviewer #2: Yes

Reviewer #3: No

5. Review Comments to the Author

**Reviewer #1:** This is an original systematic review of a relevant topic. In general the selection of articles seems to have been done rigorously leading to a comprehensive inclusion of relevant studies published in English. Nevertheless, the non-inclusion of articles on the Ardystil disease caused by spray painting among textile workers is conspicuous (and should be justified).

I have the following concerns regarding methodology:

- Please, indicate the start date for the inclusion of articles (line 84, line 169).

- line 105 (iii) confusing formulation: "inclusive of ..." to be replaced by "covering" or "dealing with ..."? what do you mean by "or organic solvents" here (as opposed to VOCs)?

- lines 174-175: Iraq is not an African country! Baghdad and Tehran are not countries but country capitals. These troublesome errors have an impact on the meta-analysis by region!

- table 4: values of FEV1 and FVC can be/are reported as percentages of predicted values (note implausible values for study #16)) or as absolute values (with corrections for sex, age and height ?); FEV1/FVC ratios are reported as percentages, most being plausibly around 80% (FEV1/FVCx100) but studies #8 and 14 have values above 90%, presumably because they were expressed as percent of a predicted ratio (not a good option!). HOW WERE THESE DIFFERENCES IN THE EXPRESSION OF SPIROMETRIC PARAMETERS TAKEN INTO ACCOUNT FOR THE QUANTITATIVE ANALYSES ? Even when using SMDs, this issue is crucial for the general credibility of the analyses and conclusions.

Other comments:

- Tables: please include the corresponding reference numbers (in the list of references) in all tables and figures.

- Table 2: in the outcome column, the text can/should be much more concise (avoid unnecessary words "symptoms like ...") and make a clearer distinction between symptoms and PFTs (perhaps using separate columns).

- Tables 3, 5 and 6: add good legends

- Table 4: see comments made above about expression of parameters; legend: ** values in liters (not litres/minute), last phrases of legend make no sense

- Figure 2, 3, 4, 5: please harmonize the styles across figures, add suitable explanatory legends, specify units as needed, check and harmonize how study authors are identified

- Discussion:

-- the (speculative) mechanistic explanations (lines 323-330) may be deleted.

-- line 392: gaps were not really identified: suggestions are:

--- why, unusually, so few studies from industrially developed countries?

--- what about prevalence of clinical pulmonary disease (asthma or COPD), what about pulmonary fibrosis?

--- what is the impact in terms of overall morbidity and mortality (proportion of painters in the population)? what is the public health impact of painting on respiratory health compared to other occupational exposures (mining, construction, ...)?

--- what about different types of paints (solvent-based vs water-based paints) and application modes (spraying vs brush)?

Language: generally OK, but some editing is needed (pulmonary function is generally singular; "the effect reduced", ...).

**Reviewer #2**: The present study is a systematic review and meta-analysis to determine pulmonary function and symptoms in paint industry workers. The review of the literature, the qualitative evaluation of the articles and the calculation of the pooled index were done using an appropriate scientific method. The presentation of results is appropriate and in the discussion section, interpretation has made based on the obtained results.

**Reviewer #3: **Major issues:

• I would suggest to remove any speculation on the obstructive disorder, including those in the conclusions, because the review did not analyze the obstructive disorder as such, but rather a SMD in FEV1/FVC in the exposed vs unexposed groups. Even despite some identified difference (through SMD), obstruction has clear and distinct diagnostic criterion which is FEV1/FVC below LLN or when LLN is not available, below 70%. Therefore, obstructive defect can only be treated a binary variable. In other words, no such variable was analyzed in the review, and only the difference between the groups in FEV1/FVC should be discussed.

• Albeit some hint on the limitations related to no adjustment for potential confounders is made in the Discussion, this still remains a serious issue. Given that there is an array of confounders for both lung function and symptoms, including sex, regular exercise (please see https://pubmed.ncbi.nlm.nih.gov/31920299/) and other exist, the discussion on that should be extended. Please speculated over no males in the analysis and even more important – no adjustment in the effect for confounders. This comment also extends to the comment for crude effects calculation below. When you summarized the overall effect in the conclusions (including the abstract), please state that all effects were crude, and the effects of confounders was omitted.

• Despite some discussion on very high heterogeneity, a serious concern remains for almost all polled analysis in this report, including those where I2 reached 99%. This is an extreme heterogeneity, and apparently more discussion is needed on the sources of heterogeneity. I suppose of these could be surprisingly low samples in selected studies, introducing very poor precision. This of other sources and discuss this extreme heterogeneity in more detail.

• Overall, conclusions in this review are constructed on cross-sectional studies, assuming poor evidence. Limitations of this design are well-known, but no temporality and other limitations should be louder pronounced in the results and even more in the discussion and conclusions. This is a very important finding.

A few further comments:

• De-abbreviate VOC at first use in the abstract and in the main text. Same relates to PFT in line 121. Furthermore, wen you have already abbreviated VOC in the beginning, use only VOC later on (line 229), and check all over the manuscript.

• Use a uniform term across the manuscript. At present, you use “pulmonary function”, “respiratory function”. Moreover, the terms “pulmonary functions” as plural is usually not used; instead, use “pulmonary function” if you chose this out of two (“pulmonary function” vs. “respiratory function”).

• Abstract: The aim in the abstract only mentions pulmonary function. But the title and results all over the manuscript also mention symptoms. Please fix this discrepancy. Results: delete “of the lungs” when you report SMD for FEV1 and FEV1/FVC. Conclusions: these conclusions are poorly focused. Please make them succinct.

• Introduction: Many statements in the Introduction do not have references, starting with the very first sentence. Is you narrate the exposure has detrimental effects, then present references that confirm the association. In lines 67-68 you refer to several studies, but cite only one. Same relates to lines 72-75.

• Methods.

- Why was the age limit 15-60 set as a selection criterion?

- Line 143: what range? Interquartile range? Moreover, what dictated the choice between reporting means and medians? Normality? Then state that.

- Line 144: who calculated ORs? Did the authors of this review did that? If so, how then 95% CIs were computed? Furthermore, if the authors of this review did the computation but not originally reported in the source articles, this assumes that all computed ORs were crude, and no adjusted for confounders was possible. If that is the case, discuss that in the limitations. This should also be clearly stated in the footer to Table 3.

- Line 145: the rationale to report SMD is not provided. Please clearly state that some studies reported actual FEV1 or FVC or the ratio, but some of them reported %predicted values. Therefore, direct comparison was not possible, and SMD calculation was needed.

• Results: Lines 168-180: state that none of included studies applied case-control or cohort design. This is a very important finding, assuming the despite quite high scores, overall the quality of evidence is quite low, since cross-sectional design has serious limitations. In addition, summarize how many studies used questionnaires for exposure assessment as opposed to measured exposure. Any studies using JEMs?

• Results: how would you explain that subgroup analysis of FEV1 and FVC with regard to smoking yielding opposite results? Is the reason very few eligible studies in the FVC analysis?

• Discussion: Normally tables and figures are not cited in the discussion. Please remove that and leave only text.

• Language needs more effort. Please proofread the text. One of the problems is sequence of tenses. Carefully check grammar. In addition, some typos (line 346: wide) are also present.

• Inclusion of only English studies is also a limitation and should be mentioned in the limitations.

6. PLOS authors have the option to publish the peer review history of their article (what does this mean?). If published, this will include your full peer review and any attached files.

Reviewer #1: No

Reviewer #2: **Yes: **Marzieh Nojomi

Reviewer #3: No

---

## [Author Response · Author response to Decision Letter 0]

22 Jun 2024

PONE-D-24-08732

Respiratory symptoms and Pulmonary function in paint industry workers: A Systematic Review and Meta-Analysis

Dear Editor in-chief,

We appreciate your consideration of our manuscript for the peer review procedure. The valuable feedback and suggestions provided by the reviewers to refine our current manuscript have been valued sincerely. To the best of our ability, we have responded to each comment with supplementary information and clarifications, while also adhering to the PlosONE guidelines for review articles. 

Response to reviewers

Thank you for your valuable insights and constructive feedback regarding our systematic review and Meta-analysis.

Reviewer #1: 

Query #1:This is an original systematic review of a relevant topic. In general the selection of articles seems to have been done rigorously leading to a comprehensive inclusion of relevant studies published in English. Nevertheless, the non-inclusion of articles on the Ardystil disease caused by spray painting among textile workers is conspicuous (and should be justified).

Response: Thank you for your thoughtful comment. Our current research is primarily focused on investigating the respiratory health effects of Volatile Organic Compounds (VOCs). Nevertheless, Ardystil syndrome is predominantly associated with the inhalation of aerosolized polyacrylic resins, (which is not a VOC) such as Acramin, which are frequently used in textile printing. This condition is mainly linked to three primary polycationic paint components: Acramin FWR (a polyurea), Acramin FWN (a polyamide-amine), and Acrafix FHN (a polyamine). Of these, Acramin FWR has been implicated in the development of interstitial lung disease among workers using the airbrushing technique. 

Hence, inclusion of such studies would fall outside the specific scope of our current research. This is the reason why they were not included in this systematic review. However, we recognize the importance of these studies and plan to undertake a separate systematic review in the near future to explore the toxicological effects of other paint components, including Acramin and similar substances.

Query #2: I have the following concerns regarding methodology: Please, indicate the start date for the inclusion of articles (line 84, line 169).

Response: Thank you for your feedback regarding the methodology. In accordance with your suggestion, we have specified the start date as "from inception". This designation ensures that our review includes all relevant research articles published from their inception up to August 2024.

Pages: 5,12 Lines: 86, 171

Query #3: line 105 (iii) confusing formulation: "inclusive of ..." to be replaced by "covering" or "dealing with ..."? what do you mean by "or organic solvents" here (as opposed to VOCs)?

Response: Thank you for your feedback. We have revised the formulation as per your suggestion, replacing "inclusive of ..." with "covering" to enhance clarity. Additionally, we have removed the phrase "present in paints or organic solvents" to avoid any ambiguity.

Page:6 Line: 106

Query #4: lines 174-175: Iraq is not an African country! Baghdad and Tehran are not countries but country capitals. These troublesome errors have an impact on the meta-analysis by region!

Response: Thank you for pointing out these errors. We apologize for the oversight. We have corrected the inaccuracies regarding Iraq, Baghdad, and Tehran, ensuring accurate representation in both the methodology and results sections. Your suggestion has been invaluable in improving the integrity of our meta-analysis by region.

Page: 12 Lines: 175-179, Table:2,6

Query #5: table 4: values of FEV1 and FVC can be/are reported as percentages of predicted values (note implausible values for study #16)) or as absolute values (with corrections for sex, age and height?); 

Response: Thank you for your observation regarding Table 4. We have addressed the issue, specifically for study #16, by reporting FEV1 and FVC values as percentages of predicted values. This adjustment ensures accuracy and consistency in our presentation of the data.

Pages: 16 Line: 239

Query #6: FEV1/FVC ratios are reported as percentages, most being plausibly around 80% (FEV1/FVCx100) but studies #8 and 14 have values above 90%, presumably because they were expressed as percent of a predicted ratio (not a good option!). HOW WERE THESE DIFFERENCES IN THE EXPRESSION OF SPIROMETRIC PARAMETERS TAKEN INTO ACCOUNT FOR THE QUANTITATIVE ANALYSES ? Even when using SMDs, this issue is crucial for the general credibility of the analyses and conclusions.

Response: As correctly stated, studies #8 and #14 presented the FEV1/FVC ratio values as predicted percentages, while the remaining studies reported them as absolute values. To maintain consistency across the studies, we standardized the mean differences of the reported FEV1/FVC ratios in our quantitative analysis. This approach allowed us to maintain a high level of consistency in both the calculation and interpretation of spirometry parameters across the entire dataset. We would like to highlight that we have computed the standardized mean difference of the exposed and unexposed groups, which were represented as either absolute values or predicted values. It is of utmost importance to ensure that the conclusion derived from our meta-analysis accurately represents the combined evidence from the included studies. We would like to extend our sincere apologies for the oversight regarding the absence of the " * " marking in study #8, which might have caused confusion.

In addition, we would like to cite an article that includes calculations of mean difference FEV1/FVC ratios expressed as a percentage of predicted values.

Jing Z, Wang X, Zhang P, Huang J, Jia Y, Zhang J, Wu H, Sun X. Effects of physical activity on lung function and quality of life in asthmatic children: An updated systematic review and meta-analysis. Frontiers in Pediatrics. 2023 Feb 8;11:1074429. 

Query #7: Tables: please include the corresponding reference numbers (in the list of references) in all tables and figures.

Response: As per the suggestion, we have included the corresponding reference numbers (in the list of references) in all tables and figures.

Query #8: Table 2: in the outcome column, the text can/should be much more concise (avoid unnecessary words "symptoms like ...") and make a clearer distinction between symptoms and PFTs (perhaps using separate columns).

Response: Thank you for your feedback regarding Table 2. We have updated the table to separate Respiratory symptoms and Pulmonary Function Tests (PFTs) into distinct columns. Additionally, we have revised the text in the outcome column to ensure conciseness, eliminating unnecessary wording such as "symptoms like ...". These changes enhance the clarity and organization of the data presented in the table.

Query #9: Tables 3, 5 and 6: add good legends

We have added good legends in the above tables based on your suggestion.

Query #10: Table 4: see comments made above about expression of parameters; legend: ** values in liters (not litres/minute), last phrases of legend make no sense

Response: The necessary modifications have been made in Table.4.

Query #11: Figure 2, 3, 4, 5: please harmonize the styles across figures, add suitable explanatory legends, specify units as needed, check and harmonize how study authors are identified

Response: We have harmonized the style across the figures, added suitable explanatory legends, specified the units as needed and harmonized the style across study authors as well.

Query #12: Discussion: the (speculative) mechanistic explanations (lines 323-330) may be deleted.

Response: We have removed the speculative mechanistic explanations from the discussion section as recommended.

Query #13: line 392: gaps were not really identified: suggestions are: why, unusually, so few studies from industrially developed countries? what about prevalence of clinical pulmonary disease (asthma or COPD), what about pulmonary fibrosis? What is the impact in terms of overall morbidity and mortality (proportion of painters in the population)? What is the public health impact of painting on respiratory health compared to other occupational exposures (mining, construction,….)? What about different types of paints (solvent-based vs water-based paints) and application modes (spraying vs brush)?

Response: We have added the identified gaps and made the necessary changes in the lines starting from 392 in accordance with the feedback. 

Page: 23,24 Lines:396 -407

Query #14: Language: generally OK, but some editing is needed (pulmonary function is generally singular; "the effect reduced", ...).

Response: We have incorporated the reviewer’s suggestions and made the necessary changes in the entire manuscript. 

Reviewer #2: The present study is a systematic review and meta-analysis to determine pulmonary function and symptoms in paint industry workers. The review of the literature, the qualitative evaluation of the articles and the calculation of the pooled index were done using an appropriate scientific method. The presentation of results is appropriate and in the discussion section, interpretation has made based on the obtained results.

Response: Thank you for your positive feedback and appreciation of our study. We are pleased that you found our systematic review and meta-analysis methodology appropriate and the presentation of results satisfactory. Your comments encourage us to continue our efforts in contributing to the field of occupational health research. 

Reviewer #3: 

Thank you for your valuable insights and constructive feedback regarding our systematic review and Meta analysis.

Major issues:

Query #1: I would suggest to remove any speculation on the obstructive disorder, including those in the conclusions, because the review did not analyze the obstructive disorder as such, but rather a SMD in FEV1/FVC in the exposed vs unexposed groups. Even despite some identified difference (through SMD), obstruction has clear and distinct diagnostic criterion which is FEV1/FVC below LLN or when LLN is not available, below 70%. Therefore, obstructive defect can only be treated a binary variable. In other words, no such variable was analyzed in the review, and only the difference between the groups in FEV1/FVC should be discussed.

Response: Thank you for your insightful perspective regarding the reporting of obstructive disorder in our review. We acknowledge that our study focused on the standardized mean difference (SMD) in FEV1/FVC between exposed and unexposed groups, rather than analyzing obstructive disorder as defined by specific diagnostic criteria. In response to your suggestion, we have removed or replaced such phrases and made the necessary adjustments accordingly.

Query #2: Albeit some hint on the limitations related to no adjustment for potential confounders is made in the Discussion, this still remains a serious issue. Given that there is an array of confounders for both lung function and symptoms, including sex, regular exercise (please see https://pubmed.ncbi.nlm.nih.gov/31920299/) and other exist, the discussion on that should be extended. 

Response: We appreciate your insightful comments regarding the limitations related to potential confounders in our study. The odds ratios mentioned in the study are the crude odds ratios calculated by the author. Unfortunately, the lack of data prevented us from taking into account potential confounders, such as age, and effect modifiers, such as exercise, in our analysis. In addition, we have made appropriate modifications in our discussion.

Pages: 23, 24 Lines:387-389, 414 -416

Query #3:Please speculated over no males in the analysis and even more important – no adjustment in the effect for confounders.

Response: We appreciate your query, it is important to note that the studies included in the analysis primarily concentrated on male populations, with only two exceptions that specifically investigated respiratory symptoms.

Pages: 22,15, 22 Lines: 369 -371,235,369,

Query #4: This comment also extends to the comment for crude effects calculation below. When you summarized the overall effect in the conclusions (including the abstract), please state that all effects were crude, and the effects of confounders was omitted.

Response: Thank you for your suggestion. We have incorporated your recommendation by explicitly stating in the abstract, results, discussion, and conclusion sections that the odds ratios reported for respiratory symptoms are crude estimates, as confounding factors were not adjusted for in our analysis.

Pages: 13,14,18,24 Lines:45, 202, 203, 208, 209, 215, 223, 286, 287, 292, 293, 294, 387-389, 414 -416.

Query #5: Despite some discussion on very high heterogeneity, a serious concern remains for almost all polled analysis in this report, including those where I2 reached 99%. This is an extreme heterogeneity, and apparently more discussion is needed on the sources of heterogeneity. I suppose these could be surprisingly low samples in selected studies, introducing very poor precision. This of other sources and discuss this extreme heterogeneity in more detail.

Response: As you rightly stated one of the probable reasons for heterogeneity has been discussed in the discussion section.

Page: 22 Lines:372 - 376

Query #6: Overall, conclusions in this review are constructed on cross-sectional studies, assuming poor evidence. Limitations of this design are well-known, but no temporality and other limitations should be louder pronounced in the results and even more in the discussion and conclusions. This is a very important finding.

Response: As per your suggestion, we have documented about the limitations of the study group in the results,discussion and in conclusion sections.

Pages: 23,24 Lines:382-387,413-414

Query #7: De-abbreviate VOC at first use in the abstract and in the main text. Same relates to PFT in line 121. Furthermore, when you have already abbreviated VOC in the beginning, use only VOC later on (line 229), and check all over the manuscript.

Response: The abbreviation for VOCs, PFT have been expanded upon its first use, both in the abstract and in the main text. Additionally, we have incorporated the abbreviated form throughout the manuscript based on your valuable suggestions.

Pages:3,4,6 Lines:36,43,59,121

Query #8: Use a uniform term across the manuscript. At present, you use “pulmonary function”, “respiratory function”. Moreover, the terms “pulmonary functions” as plural is usually not used; instead, use “pulmonary function” if you chose this out of two (“pulmonary function” vs. “respiratory function”).

Response: As per your recommendation, respiratory function has been uniformly changed to pulmonary function. 

Query #9: Abstract: The aim in the abstract only mentions pulmonary function. But the title and results all over the manuscript also mention symptoms. 

Response: We have included the respiratory symptoms under “Aim” in the abstract which was missed inadvertently earlier.

Query #10: Please fix this discrepancy. Results: delete “of the lungs” when you report SMD for FEV1 and FEV1/FVC. 

Response: We have deleted the phrase "of the lungs" as per your request.

Query #11: Conclusions: these conclusions are poorly focused. Please make them succinct.

As per your suggestion, we have made the necessary changes in conclusion

Page: 24 Lines: 409 -425

Query #12: Introduction: Many statements in the Introduction do not have references, starting with the very first sentence. Is you narrate the exposure has detrimental effects, then present references that confirm the association. 

Response: We have added the references in the “Introduction” section as suggested.

Page: 24 Lines: 409 -425

Query #13: In lines 67-68 you refer to several studies, but cite only one. Same relates to lines 72-75.

 Response: We have cited as per your suggestion. Apologies for missing the same earlier.

Query #14: Why was the age limit 15-60 set as a selection criterion?

The age limit of 15-60 years was chosen as the 15 to 60 age range encompasses the majority of the working-age population, during which lung function tends to stabilize and remain relatively consiste

---

## [Editor Report · Decision Letter 1]

9 Jul 2024

PONE-D-24-08732R1Respiratory symptoms and Pulmonary function in paint industry workers: A Systematic Review and Meta-AnalysisPLOS ONE

Dear Dr. Johnson,

Thank you for submitting your manuscript to PLOS ONE. After careful consideration, we feel that it has merit but does not fully meet PLOS ONE’s publication criteria as it currently stands. Therefore, we invite you to submit a revised version of the manuscript that addresses the points raised during the review process.

 Please be aware of the suggestions provided by the Reviewers, especially those from the Reviewer 1 who still has some specific requests. I strongly recommend to make all the suggested changes to improve the manuscript.

We look forward to receiving your revised manuscript.

Kind regards,

Giulia Squillacioti

Academic Editor

PLOS ONE
---

## [Author Response · Author response to Decision Letter 1]

29 Aug 2024

PONE-D-24-08732

Respiratory symptoms and Pulmonary function in paint industry workers: A Systematic Review and Meta-Analysis

Dear Editor in-chief,

We appreciate your consideration of our work for the peer review process. Thank you for providing us with valuable suggestions. We truly appreciate your input. 

#Comment 1

Please be aware of the suggestions provided by the Reviewers, especially those from the Reviewer 1 who still has some specific requests. I strongly recommend to make all the suggested changes to improve the manuscript.

Thank you for taking the time to review our work. Your effort and input are greatly appreciated. We have carefully reviewed the suggestions given by reviewer 1 and have made the required revisions to the best of our ability. We have provided justifications and citations for your comments, particularly for queries 1, 6, 7, 10, 12 and 14 of reviewer 1. We have also thoroughly reviewed all the other comments by reviewer 1 to the best of our ability.

Query #1:This is an original systematic review of a relevant topic. In general the selection of articles seems to have been done rigorously leading to a comprehensive inclusion of relevant studies published in English. Nevertheless, the non-inclusion of articles on the Ardystil disease caused by spray painting among textile workers is conspicuous (and should be justified).

Response: Thank you for your thoughtful comment. Our current research is primarily focused on investigating the respiratory health effects of Volatile Organic Compounds (VOCs). Nevertheless, Ardystil syndrome is predominantly associated with the inhalation of aerosolized polyacrylic resins, (which is not a VOC) such as Acramin, which are frequently used in textile printing. This condition is mainly linked to three primary polycationic paint components: Acramin FWR (a polyurea), Acramin FWN (a polyamide-amine), and Acrafix FHN (a polyamine). Of these, Acramin FWR has been implicated in the development of interstitial lung disease among workers using the airbrushing technique. 

Hence, inclusion of such studies would fall outside the specific scope of our current systematic review. In addition, we have clearly mentioned in the exclusion criteria that studies dealing with major exposures other than VOCs were excluded. However, we recognize the importance of these studies and plan to undertake a separate systematic review in the near future to explore the toxicological effects of other paint components, including Acramin and similar substances.

Query #2: I have the following concerns regarding methodology: Please, indicate the start date for the inclusion of articles (line 84, line 169).

Response: Thank you for your feedback regarding the methodology. In accordance with your suggestion, we have specified the start date as "from inception". This designation ensures that our review includes all relevant research articles published from their inception up to August 2024.

Pages: 5, 12; Lines: 86, 171

Query #3: line 105 (iii) confusing formulation: "inclusive of ..." to be replaced by "covering" or "dealing with ..."? what do you mean by "or organic solvents" here (as opposed to VOCs)?

Response: Thank you for your feedback. We have revised the formulation as per your suggestion, replacing "inclusive of ..." with "covering" to enhance clarity. Additionally, we have removed the phrase "or organic solvents" to avoid any ambiguity.

Page: 6; Line: 106

Query #4: lines 174-175: Iraq is not an African country! Baghdad and Tehran are not countries but country capitals. These troublesome errors have an impact on the meta-analysis by region!

Response: Thank you for pointing out these errors. We apologize for the oversight. We have corrected the inaccuracies regarding Iraq, Baghdad, and Tehran, ensuring accurate representation in both the methodology and results sections. Your suggestion has been invaluable in improving the integrity of our meta-analysis by region.

Page: 12; Lines: 175-179, Table: 2, 6

Query #5: table 4: values of FEV1 and FVC can be/are reported as percentages of predicted values (note implausible values for study #16)) or as absolute values (with corrections for sex, age and height?)

Response: Thank you for your observation regarding Table 4. We have made the necessary corrections in study #16, and reported FEV1 and FVC values as percentages of predicted values. This adjustment ensures accuracy and consistency in our presentation of the data.

Page: 16; Table: 4 

Query #6: FEV1/FVC ratios are reported as percentages, most being plausibly around 80% (FEV1/FVCx100) but studies #8 and 14 have values above 90%, presumably because they were expressed as percent of a predicted ratio (not a good option!). 

As correctly stated, studies #8 and #14 presented the FEV1/FVC ratio values as predicted percentages, while the remaining studies reported them as absolute values. We would like to extend our sincere apologies for the oversight regarding the absence of the " * " marking in study #8, which might have caused confusion.

Page: 15, 16; Table: 4 

Query #7: HOW WERE THESE DIFFERENCES IN THE EXPRESSION OF SPIROMETRIC PARAMETERS TAKEN INTO ACCOUNT FOR THE QUANTITATIVE ANALYSES? Even when using SMDs, this issue is crucial for the general credibility of the analyses and conclusions.

Response: To maintain consistency across the studies, we performed standardized the mean differences (SMD) of the reported FEV1/FVC ratios in our quantitative analysis. i.e., we have computed the SMD of the exposed and unexposed groups, which were represented as either absolute or predicted values. It is important to note that the value of individual studies remains consistent regardless of the expressions of the spirometric parameters (absolute and predicted values). For better understanding, we performed a separate analysis in which we have categorized the studies into two groups: those studies that evaluated FEV1/FVC as an absolute value (Ref Fig 1) and those that evaluated predicted values (Ref Fig 2). It is worth noting that while comparing with the combined SMD (Ref Fig 3), the values of each study remains to be same. This demonstrates the ability to merge both expressions, resulting in improved clarity and analysis. It is of utmost importance to ensure that the conclusion derived from our meta-analysis accurately represents the combined evidence from the included studies.

We would like to cite few references regarding calculation of SMD’s for further clarification

• Andrade C. Mean difference, standardized mean difference (SMD), and their use in meta-analysis: as simple as it gets. The Journal of clinical psychiatry. 2020 Sep 22;81(5):11349.

• Gallardo‐Gómez D, Richardson R, Dwan K. Standardized mean differences in meta‐analysis: A tutorial. Cochrane Evidence Synthesis and Methods. 2024 Mar;2(3):e12047.

Ref Fig 1(FEV1/FVC (absolute),Ref Fig 2 (predicted %),Ref Fig 3 (combined) : Please find attached pictures in the response to reviewers)

Query #8: Tables: please include the corresponding reference numbers (in the list of references) in all tables and figures.

Response: As per the suggestion, we have included the corresponding reference numbers (in the list of references) in all tables and figures.

Query #9: Table 2: in the outcome column, the text can/should be much more concise (avoid unnecessary words "symptoms like ...") and make a clearer distinction between symptoms and PFTs (perhaps using separate columns).

Response: Thank you for your feedback regarding Table 2. We have updated the table to separate Respiratory symptoms and Pulmonary Function Tests (PFTs) into distinct columns. Additionally, we have revised the text in the outcome column to ensure conciseness, eliminating unnecessary wording such as "symptoms like ...". These changes enhance the clarity and organization of the data presented in the table.

Query #10: Tables 3, 5 and 6: add good legends

We reworked on it and added suitable legends to the above tables based on your valuable suggestion.

Query #11: Table 4: see comments made above about expression of parameters; legend: ** values in liters (not litres/minute), last phrases of legend make no sense

Response: The necessary modifications have been made in Table.4.

Query #12: Figure 2, 3, 4, 5: please harmonize the styles across figures, add suitable explanatory legends, specify units as needed, check and harmonize how study authors are identified

Response: We have harmonized the style across the figures, added suitable explanatory legends, specified the units as needed and harmonized the style across study authors as well.

Query #13: Discussion: the (speculative) mechanistic explanations (lines 323-330) may be deleted.

Response: We have removed the speculative mechanistic explanations mentioned in lines 323 – 330 from the discussion section as recommended.

Query #14: line 392: gaps were not really identified: suggestions are: why, unusually, so few studies from industrially developed countries? what about prevalence of clinical pulmonary disease (asthma or COPD), what about pulmonary fibrosis? What is the impact in terms of overall morbidity and mortality (proportion of painters in the population)? What is the public health impact of painting on respiratory health compared to other occupational exposures (mining, construction,….)? What about different types of paints (solvent-based vs water-based paints) and application modes (spraying vs brush)?

Response: We appreciate your valuable suggestions. We have incorporated the following sentences to address the gaps and have made the appropriate revisions in the discussion. 

“It can be inferred from the current review that paint industry workers exposed to VOCs exhibit greater respiratory health impacts such as a decline in pulmonary function. Furthermore, this review served as a needs assessment, allowing us to identify gaps in the current knowledge of pulmonary function among painters. Limited research assessed the pulmonary function in developed countries and respiratory symptoms in Asian countries. There is a dearth of sufficient evidence about the influence of VOCs on painters working in unorganized sectors such as construction and paint manufacturing, as compared to organized sectors such as the automobile industry. The prevalence of respiratory ailments, including COPD and pulmonary fibrosis, among painters is not well-documented. However, other meta-analyses have reported that the pooled risk ratio of studies that evaluated the impact of VOC exposure on asthma was higher, particularly for exposures to benzene, toluene, and p-dichlorobenzene (50). A detailed assessment is needed to assess the impacts of solvent-based paints against water-based paints, as well as the differences between spray painting and brushing/rolling methods. Therefore, it is crucial to do further study in this topic to ascertain the actual impact.” 

Page: 23,24 Lines:398 -407.

Query #15: Language: generally OK, but some editing is needed (pulmonary function is generally singular; "the effect reduced", ...).

Response: We have incorporated the reviewer’s suggestions and made the necessary changes in the entire manuscript. 

Journal Requirements:

After a thorough review of the entire reference list, we did not find any articles that have been retracted. However, some of the listed references were not cited properly in the bibliography, so we have made the necessary changes. Furthermore, kindly inform us if any references require correction or revision. Your feedback is greatly appreciated and we will make the necessary adjustments accordingly.

Ref No 32 Aribo.E.O, Antai A.B. Lung function parameters in spray painters in Calabar,Nigeria.Ann Biol.Res. 2014 Dec ; 5(11):32-5.

Ref No.42 Mukhtar M, Gotmare P. Health hazard and occupational safety challenges for Unorganized sector workers in India. J Contemp Issues Bus Gov. 2021 Jun Feb 9;27 (1):3210-21.

2. While revising your submission, please upload your figure files to the Preflight Analysis and Conversion Engine (PACE) digital diagnostic tool, https://pacev2.apexcovantage.com/. PACE helps ensure that figures meet PLOS requirements. 

As per your suggestion we have upload our figures files to the Preflight Analysis and Conversion Engine (PACE) digital diagnostic tool and converted the figures to meet PLOS requirements.

---

## [Decision Letter · Decision Letter 2]

23 Sep 2024

PONE-D-24-08732R2Respiratory symptoms and Pulmonary function in paint industry workers: A Systematic Review and Meta-AnalysisPLOS ONE

Dear Dr. Johnson,

Thank you for submitting your manuscript to PLOS ONE. After careful consideration, we feel that it has merit but does not fully meet PLOS ONE’s publication criteria as it currently stands. Therefore, we invite you to submit a revised version of the manuscript that addresses the points raised during the review process.

To date, only a few comments are still pending. Please comply with the requests provided by the Reviewer #1 to improve the manuscript still further.

We look forward to receiving your revised manuscript.

Kind regards,

Giulia Squillacioti

Academic Editor

PLOS ONE

Journal Requirements:

Reviewers' comments:

Reviewer's Responses to Questions

**Comments to the Author**

1. If the authors have adequately addressed your comments raised in a previous round of review and you feel that this manuscript is now acceptable for publication, you may indicate that here to bypass the “Comments to the Author” section, enter your conflict of interest statement in the “Confidential to Editor” section, and submit your "Accept" recommendation.

Reviewer #1: (No Response)

Reviewer #3: All comments have been addressed

2. Is the manuscript technically sound, and do the data support the conclusions?

Reviewer #1: Yes

Reviewer #3: Yes

3. Has the statistical analysis been performed appropriately and rigorously? 

Reviewer #1: Yes

Reviewer #3: Yes

4. Have the authors made all data underlying the findings in their manuscript fully available?

Reviewer #1: Yes

Reviewer #3: Yes

5. Is the manuscript presented in an intelligible fashion and written in standard English?

Reviewer #1: Yes

Reviewer #3: Yes

6. Review Comments to the Author

Reviewer #1: The authors have satisfactorily addressed my comments/questions.

I have only few editorial issues to mention:

- Title: consider specifying in the title that the review focuses on exposure to VOLATILE ORGANIC COMPOUNDS in painters

- line 179: "America", specify North America or United States of America

- lines 382-3: clarify sentence

- line 385: the present review HAS certain limitations

- line 388: I suggest replacing "increase" and "decrease ..." by "excess" and "lower mean pulmonary function" compared to controls ("increase" and "decrease" suggest a temporal process that cross-sectional studies cannot assess)

- line 398: "greater respiratory impact": greater than what or who ? I suggest to replace this by "adverse respiratory health impacts"

- line 413: see comment above: replace "notable increase" and "decline" by "higher prevalence" and "lower average", respectively

- references: some references mention first names in full:16, 22, 26, 38; reference 44 has strange author names.

Reviewer #3: None. I think my comments ahve been addressed at an earlier stage of the review. I do not see rebuttal to my primary comments in the current version.

7. PLOS authors have the option to publish the peer review history of their article (what does this mean?). If published, this will include your full peer review and any attached files.

Reviewer #1: No

Reviewer #3: No

---

## [Author Response · Author response to Decision Letter 2]

12 Nov 2024

Reviewer #1: The authors have satisfactorily addressed my comments/questions. I have only few editorial issues to mention:

We appreciate your scholarly peer review process and the time you have spent to share your thoughtful suggestions with us. Your input is greatly appreciated. 

#Comment 1

Title: consider specifying in the title that the review focuses on exposure to VOLATILE ORGANIC COMPOUNDS in painters

Thank you for your valuable comment. As per your suggestions, we have specified VOC exposure in the title

New Title: Respiratory symptoms and Pulmonary function in paint industry workers exposed to Volatile Organic Compounds: A Systematic Review and Meta-Analysis

#Comment 2: line 179: "America", specify North America or United States of America

Thank you for your valuable suggestions. We have specified America as United States of America

#Comment 3: lines 382-3: clarify sentence

As per your suggestion we have rephrased the sentence for better clarity

Rephrased sentence: Although publication bias existed studies that assessed the pulmonary function parameters, Fail-safe (sensitivity) analysis showed that a substantially greater number of studies with null values would be required to negate the present findings.

#Comment 4: line 385: the present review HAS certain limitations

Thank you for identifying the typo error

#Comment 5: line 388: I suggest replacing "increase" and "decrease ..." by "excess" and "lower mean pulmonary function" compared to controls ("increase" and "decrease" suggest a temporal process that cross-sectional studies cannot assess)

Following your suggestion, we have changed the two terms.

#Comment 6: line 398: "greater respiratory impact": greater than what or who ? I suggest to replace this by "adverse respiratory health impacts"

As per your suggestion we have replaced the words

#Comment 7: line 413: see comment above: replace "notable increase" and "decline" by "higher prevalence" and "lower average", respectively

Following your suggestion, we have changed the two terms.

#Comment 8: references: some references mention first names in full: 16, 22, 26, 38; reference 44 has strange author names.

Thank you for pointing out the errors. We have made the necessary changes according to your suggestion.

---

## [Editor Report · Decision Letter 3]

26 Nov 2024

Respiratory symptoms and Pulmonary function in paint industry workers exposed to Volatile Organic Compounds: A Systematic Review and Meta-Analysis

PONE-D-24-08732R3

Dear Dr. Johnson,

We’re pleased to inform you that your manuscript has been judged scientifically suitable for publication and will be formally accepted for publication once it meets all outstanding technical requirements.

Kind regards,

Giulia Squillacioti

Academic Editor

PLOS ONE
---

## [Editor Report · Acceptance letter]

8 Dec 2024

PONE-D-24-08732R3 

PLOS ONE

Dear Dr. Johnson, 

I'm pleased to inform you that your manuscript has been deemed suitable for publication in PLOS ONE. Congratulations! Your manuscript is now being handed over to our production team.

Kind regards, 

on behalf of

Dr. Giulia Squillacioti 

Academic Editor

PLOS ONE